# TANGO: Time-reversal Latent GraphODE for Multi-Agent Dynamical Systems

## Abstract

Learning complex multi-agent system dynamics from data is crucial across many domains, such as in physical simulations and material modeling.Extended from purely data-driven approaches, existing physics-informed approaches such as Hamiltonian Neural Network strictly follow energy conservation law to introduce inductive bias, making their learning more sample efficiently. However, many real-world systems do not strictly conserve energy, such as spring systems with frictions. Recognizing this, we turn our attention to a broader physical principle: *Time-Reversal Symmetry*, which depicts that the dynamics of a system shall remain invariant when traversed back over time. It still helps to preserve energies for conservative systems and in the meanwhile, serves as a strong inductive bias for non-conservative, reversible systems. To inject such inductive bias, in this paper, we propose a simple-yet-effective self-supervised regularization term as a soft constraint that aligns the forward and backward trajectories predicted by a continuous graph neural network-based ordinary differential equation (GraphODE). It effectively imposes time-reversal symmetry to enable more accurate model predictions across a wider range of dynamical systems under classical mechanics. In addition, we further provide theoretical analysis to show that our regularization essentially minimizes higher-order Taylor expansion terms during the ODE integration steps, which enables our model to be more noise-tolerant and even applicable to irreversible systems. Experimental results on a variety of physical systems demonstrate the effectiveness of our proposed method. Particularly, it achieves an MSE improvement of 11.5 % on a challenging chaotic triple-pendulum systems[1].

## 1 Introduction

Multi-agent dynamical systems, spanning applications from physical simulations (Battaglia et al., 2016; Kipf et al., 2018; Wang et al., 2020) to robotic control (Li et al., 2022; Gu et al., 2017), are challenging to model due to intricate and dynamic inter-agent interactions. Traditional simulators can be very time-consuming and require domain knowledge of the underlying dynamics, which are often unknown (Sanchez-Gonzalez et al., 2020; Pfaff et al., 2021). Therefore, directly learning a neural simulator from the observational data becomes an attractive alternative. A popular line of research involves using GraphODEs (Huang et al., 2020; Luo et al., 2023; Zang & Wang, 2020), where Graph Neural Networks (GNNs) serve to learn the time integration of the ordinary differential equations(ODEs), for continuous pairwise interactions among agents. Compared with discrete GNN methods (Kipf et al., 2018; Sanchez-Gonzalez et al., 2020), GraphODEs show superior performance in long-range predictions and can handle irregular and partial observations (Jiang et al., 2023).

However, the intricate nature of multi-agent systems often necessitates vast amounts of training data. Vanilla data-driven neural simulators trained on limited datasets tend to be less generalizable, and can violate physical properties of a system such as energy conservation. As depicted in Figure 1 (a.1), the learned energy curve of a baseline model (LG-ODE) (Huang et al., 2020) is prone to explosion, even though the ground-truth energy remains constant. One promising strategy to mitigate this data dependency is to incorporate physical inductive biases (Raissi et al., 2019; Cranmer et al., 2020). Existing works like Hamiltonian- Neural Nets and ODE (Greydanus et al., 2019; Sanchez-Gonzalez et al., 2019) strictly enforce the energy conservation law, leading to more accurate pre-

---

[1]Code implementation can be found at here.

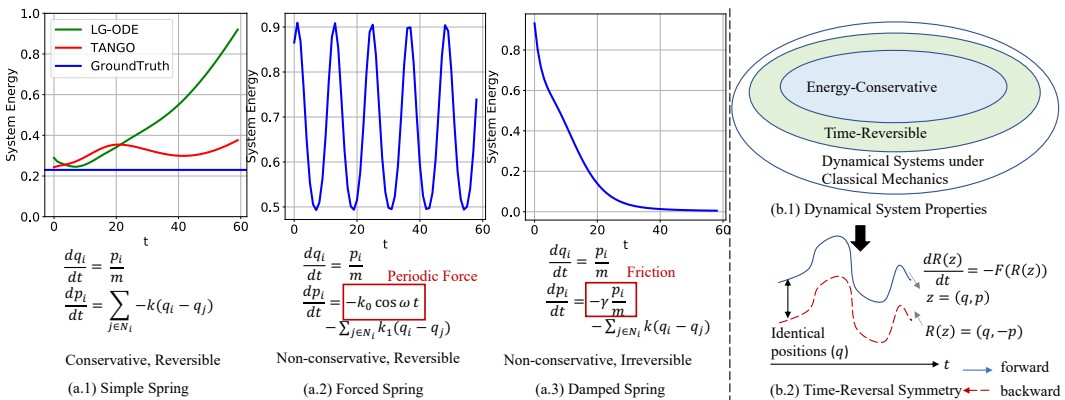

Figure 1: (a) Three $n$-body spring systems characterized by their energy conservation and time reversal properties. $p, q, m$ denote momentum, position and mass, respectively. Proof of energy conservation and time reversal for these systems can be found in Appendix B (b) Classification of classical mechanical systems based on (Tolman, 1938; Lamb & Roberts, 1998)

dictions for some systems under classical mechanics, especially in data-scarce situations. However, not all real-world systems adhere to strict energy conservation, especially those that have interaction with external environments, i.e. non-isolated systems, such as $n$-body spring systems with periodic external forces or frictions shown in Figure 1 (a.2) and (a.3). For these systems, applying strict energy conservation constraint proposed by Greydanus et al. (2019); Sanchez-Gonzalez et al. (2019) could lead to inferior performance. As shown in Figure 1(b.1), for classical and deterministic mechanics such as Newtonian mechanics, energy-conservative systems also obey time-reversal symmetry (Tolman, 1938). On the other hand, we note that the time-reversible systems also include non-conservative systems such as Stokes flow (Pozrikidis, 2001), which also has vital applications in the real world (Kim & Karrila, 2013). Therefore, by ensuring that the system's dynamics is approximately invariant under time reversal, we can enforce neural simulators to generate dynamical systems that are more realistic, paving the way for more efficient and physically coherent dynamical system modeling. In light of these observations, we pivot towards a broader physical principle: *Time-Reversal Symmetry*, which posits that a system's dynamics should remain invariant when time is reversed (Lamb & Roberts, 1998).

To incorporate such time-reversal inductive bias, we propose a simple-yet-effective self-supervised regularization term as a soft constraint that aligns forward and backward trajectories predicted by our model, which has GraphODE as its backbone. We name our model TANGO: **T**ime-Reversal L**at**e**n**t **G**raph **O**DE, which learns the system dynamics in the latent space. This time-reversal loss effectively imposes time-reversal symmetry to enable more accurate model predictions across a wider range of systems under classical mechanics. Besides its physical implication on benefiting learning *reversible* systems, we also empirically observe that the time-reversal loss in general helps to learn *irreversible* systems. Through theoretical analysis, we prove that from the numerical aspect, the time-reversal loss actually minimizes higher-order Taylor expansion terms during the ODE integration steps, which enables our model to be more noise-tolerable and even applicable to irreversible systems. Therefore, TANGO has the flexibility to be applied to a wide range of dynamical systems without requiring the systems to be strictly energy-conservative or time-reversible. We conducted systematic experiments over four simulated datasets. Experimental results verify the effectiveness of TANGO in learning system dynamics more accurately with less observational data.

The primary contributions of this paper can be summarized as follows:

- We propose TANGO, a GraphODE model that incorporates time-reversal symmetry as a soft constraint and adeptly handles both energy-conservative and non-conservative systems.
- We theoretically explain why the proposed time-reversal symmetry loss could in general help learn more fine-grained and long-context system dynamics from the numerical aspect.

- Our method achieves state-of-the-art results in multi-agent physical simulations. Particularly, it achieves an MSE improvement of 11.5 % on trajectory predictions on a challenging chaotic triple-pendulum system.

## 2    PRELIMINARIES AND RELATED WORKS

We represent a multi-agent dynamical system as a graph $\mathcal{G} = (\mathcal{V}, \mathcal{E})$, where $\mathcal{V}$ denotes the node set of $N$ agents[2] and $\mathcal{E}$ denotes the set of edges representing their physical interactions, which for simplicity we assumed to be static over time. We denote $\boldsymbol{X}(t) \in \mathbb{R}^{N \times d}$ as the feature matrix at timestamp $t$ for all agents, with $d$ as the dimension of features. Model input consists of trajectories of such feature matrices over $K$ timestamps $X(t_{1:K}) = \{\boldsymbol{X}(t_1), \boldsymbol{X}(t_2), \dots, \boldsymbol{X}(t_K)\}$ and the graph structure input $\mathcal{G} = (\mathcal{V}, \mathcal{E})$. Note that the timestamps $t_1, t_2 \cdots t_K$ can have non-uniform intervals and be of any continuous values. Our goal is to learn a neural simulator $f_\theta(\cdot) : X(t_{1:K}) \rightarrow Y(t_{K+1:T})$, which predicts node dynamics $\boldsymbol{Y}(t)$ in the future based on observations. We use $\boldsymbol{y}_i(t)$ to denote the targeted dynamic vector of agent $i$ at time $t$. In some cases when we are only predicting system feature trajectories, $\boldsymbol{Y}(\cdot) \equiv \boldsymbol{X}(\cdot)$.

### 2.1    GRAPHODE FOR MULTI-AGENT DYNAMICAL SYSTEMS

Graph Neural Networks (GNNs) have been widely used to model multi-agent dynamical systems which approximate pair-wise node (agent) interaction through message passing to impose strong inductive bias. The majority of them are discrete models, which learn a fixed-step state transition function such as to predict trajectories from timestamp $t$ to timestamp $t + 1$. However, discrete models (Battaglia et al., 2016; Kipf et al., 2018; Sanchez-Gonzalez et al., 2020) have two main limitations: (1) they are not able to adequately model systems that are continuous in nature such as $n$-body spring systems. (2) they cannot deal with irregular-observed systems, where the observations for different agents are not temporally aligned and can happen at arbitrary time intervals.

Recently, researchers propose GraphODE models (Poli et al., 2019; Huang et al., 2020; Zang & Wang, 2020; Luo et al., 2023; Wen et al., 2022) which describe multi-agent dynamical systems by a series of ODEs in a continuous manner. Specifically, they employ GNN as the ODE function and learn it in a data-driven way, making GraphODEs flexible to model a wide range of real-worldsystems (Chen et al., 2018; Rubanova et al., 2019; Huang et al., 2023). The state evolution can be described as: $\dot{\boldsymbol{z}}_i(t) := \frac{d\boldsymbol{z}_i(t)}{dt} = g(\boldsymbol{z}_1(t), \boldsymbol{z}_2(t) \cdots \boldsymbol{z}_N(t))$, where $\boldsymbol{z}_i(t) \in \mathbb{R}^d$ denotes the latent state variable for agent $i$ at timestamp $t$ and $g$ denotes the message passing function that drives the system to move forward. GraphODEs have been shown to achieve superior performance, especially in long-range predictions and can handle data irregularity issues. They usually follow an encoder-processor-decoder architecture, where an encoder first computes the latent initial states $\boldsymbol{z}_1(0), \cdots \boldsymbol{z}_N(0)$ for all agents simultaneously based on their historical observations as in Eqn 1.

$$\boldsymbol{z}_1(0), \boldsymbol{z}_2(0), ..., \boldsymbol{z}_N(0) = f_{\text{ENC}}\big(X(t_{1:K}), \mathcal{G}\big) \tag{1}$$

Then the GNN-based ODE predicts the latent trajectories starting from the learned initial states. Given the initial states $\boldsymbol{z}_1(0), \cdots \boldsymbol{z}_N(0)$ for all agents and the ODE, we can compute the latent state $\boldsymbol{z}_i(T)$ at any desired time using a numerical ODE solver such as Runge-Kuttais (Schober et al., 2019) as shown in Eqn 2.

$$\boldsymbol{z}_i(T) = \text{ODE-Solver}\big(g, [\boldsymbol{z}_1(0), \boldsymbol{z}_2(0)...\boldsymbol{z}_N(0)], t = T)\big) = \boldsymbol{z}_i(0) + \int_{t=0}^{T} g(\boldsymbol{z}_1(t), \boldsymbol{z}_2(t) \cdots \boldsymbol{z}_N(t)) \, dt \tag{2}$$

Finally, a decoder extracts the predicted dynamics $\hat{\boldsymbol{y}}_i(1), \hat{\boldsymbol{y}}_i(2), ..., \hat{\boldsymbol{y}}_i(T)$ based on the trajectory of latent states $\boldsymbol{z}_i(1), \boldsymbol{z}_i(2), ..., \boldsymbol{z}_i(T)$.

$$\hat{\boldsymbol{y}}_i(t) = f_{\text{DEC}}(\boldsymbol{z}_i(t)) \tag{3}$$

However, vanilla GraphODEs can violate physical properties of a system such as energy conservation, resulting in unrealistic predictions. We therefore propose to inject physical inductive bias to make more accurate predictions.

---

[2]Following (Kipf et al., 2018), we use "agents" to denote "objects" in dynamical systems, which is different from "intelligent agent" in AI.

## 2.2 TIME-REVERSAL SYMMETRY

A system is said to have *Time-Reversal Symmetry* if its dynamics remain the same when the flow of time is reversed (Noether, 1971). Formally, let's consider a multi-agent dynamical system described in the matrix form $\frac{d\boldsymbol{Z}(t)}{dt} = F(\boldsymbol{Z}(t))$, where $\boldsymbol{Z}(t) \in \mathbb{R}^{N \times d}$ is the time-varying state matrix of all agents. The system is said to follow the *Time-Reversal Symmetry* if it satisfies

$$\frac{dR(\boldsymbol{Z}(t))}{dt} = -F(R(\boldsymbol{Z}(t)), \tag{4}$$

where $R(\cdot)$ is a reversing operator[3].

Time-Reversal Symmetry indicates that after the reversing operation $R(\cdot)$, the gradient of any point of the trajectory $Z(t)$ will be reversed (in the opposite direction). For example, considering the reversed trajectory of a flying ball, the velocity (i.e., the derivative of position with respect to time) at each position is the opposite.

Now we introduce a time evolution operator $\phi_\tau : \boldsymbol{Z}(t) \mapsto \phi_\tau(\boldsymbol{Z}(t)) = \boldsymbol{Z}(t + \tau)$ for arbitrary $t, \tau \in \mathbb{R}$, which satisfies $\phi_{\tau_1} \circ \phi_{\tau_2} = \phi_{\tau_1 + \tau_2}$ for all $\tau_1, \tau_2 \in \mathbb{R}^4$. The time evolution operator helps us to move forward or backward through time, thus forming a trajectory. Based on Lamb & Roberts (1998), we can integrate Eqn. 4 and have:

$$R \circ \phi_t = \phi_{-t} \circ R = \phi_t^{-1} \circ R, \tag{5}$$

which means that moving forward $t$ steps and then turning backward is equivalent to firstly turning backward and then moving to the other direction $t$ steps. Eqn. 5 has been widely used to describe time-reversal systems in existing literature (Huh et al., 2020; Valperga et al., 2022; Roberts & Quispel, 1992). Nevertheless, we propose the following lemma, which is more intuitive to understand and more straightforward to guide the design of our time-reversal regularizer.

**Lemma 1.** *Eqn 5 is equivalent to $R \circ \phi_t \circ R \circ \phi_t = I$, where $I$ denotes identity mapping.*

Lemma 1 means if we move $t$ steps forward and then turn backward and move for $t$ more steps, it shall restore back to the same state. The proof of the lemma is in Appendix A.1. It can be understood as rewinding a video to the very beginning.

*Time-Reversal Symmetry* is one of the fundamental symmetry properties in many physical systems, especially in classical mechanics (Lamb & Roberts, 1998; Huh et al., 2020). However, some systems may not strictly obey the time-reversal properties due to situations such as time-varying external forces, internal friction, and underlying stochastic dynamics. Therefore, a desired model shall be able to flexibly inject time-reversal symmetry as a soft constraint, so as to cover a wider range of real-world dynamical systems such as the three spring systems depicted in Figure 1 (a).

One prior work TRS-ODE (Huh et al., 2020) also injects time-reversal symmetry into neural networks as a soft constraint based on Eqn 5. However, they cannot model multi-agent interactions and work only for fully observable systems (with initial states as model input), while our method based on GraphODE is well suited for modeling multi-agent and irregularly observed systems. We also illustrate our implementation to achieve time-reversal symmetry can have a lower maximum error compared to their implementation in Appendix A.3, supported by empirical experiments in Sec. 4.2.

Note that there are some literature discussing reversible neural networks Chang et al. (2018); Liu et al. (2019). However, they do not resolve the time-reversal symmetry problem that we're studying. We highlight the detailed discussions in Appendix F.

## 3 METHOD

To incorporate *Time-Reversal Symmetry* in modeling multi-agent dynamical systems, we propose TANGO, a novel GraphODE framework with a flexible regularization based on Lemma 1. According to Lemma 1, we align the forward trajectory $\boldsymbol{z}^{\text{fwd}}(t)$ and backward reversed trajectory $\boldsymbol{z}^{\text{rev}}(T-t)$

---

[3]Taking a Hamiltonian system $(\boldsymbol{q}, \boldsymbol{p}, t)$ as an example, where $\boldsymbol{q}, \boldsymbol{p}$ denote position and momentum, the reversing operator $R : (\boldsymbol{q}, \boldsymbol{p}, t) \mapsto (\boldsymbol{q}, -\boldsymbol{p}, -t)$ makes the momentum reverse and traverse back over time. (Lamb & Roberts, 1998).

[4]$\circ$ denotes composition

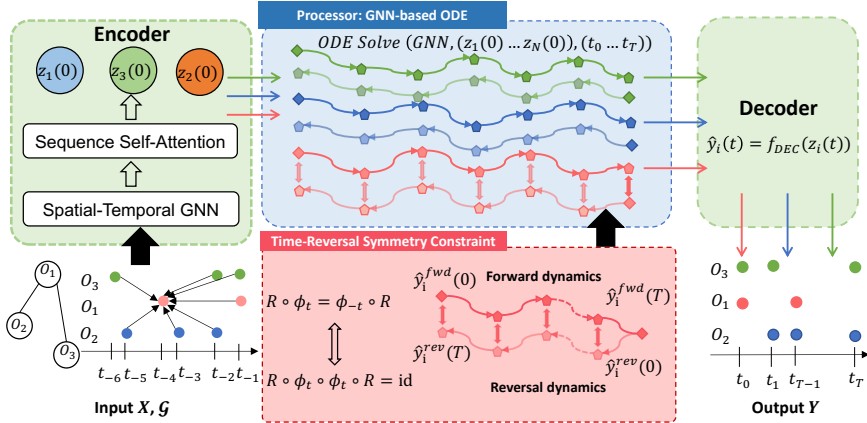

Figure 2: Overall framework of TANGO

predicted by our model, where the backward trajectory starts from the ending state of the forward one, traversed back over time. Here $t$ denotes the relative time index of the trajectory's starting point and $T$ denotes the total trajectory length. In our design, we use GraphODE to predict the above forward and reversed trajectories. The difference of the two trajectories at time $t$ and $T - t$ should be small, and thus is used as a regularizer. The weight of the regularization is also adjustable to adapt different systems. The overall framework is depicted in Figure 2.

## 3.1 TIME-REVERSAL SYMMETRY LOSS AND OVERALL TRAINING

**Forward Trajectory Prediction and Reconstruction Loss.** We utilize the GraphODE framework described in Sec.2.1 to model multi-agent dynamical systems, which follows the encoder-processor-decoder architecture. Specifically, we first utilize a Transformer-based spatial-temporal GNN described in Huang et al. (2020) as our encoder $f_{\text{ENC}}(\cdot)$ to compute the initial states from observed trajectories. The implementation details of the encoder can be found in Appendix D.1. We then utilize the GNN operator described in Kipf et al. (2018) as our ODE function $g(\cdot)$, which drives the system to move forward and output the forward trajectories for latent states $\boldsymbol{z}_i^{\text{fwd}}(t)$ at each continuous time $t$ and each agent $i$. Finally, we employ a Multilayer Perceptron (MLP) as a decoder to predict output dynamics based on the latent states. We summarize the whole procedure as:

$$\dot{\boldsymbol{z}}_i^{\text{fwd}}(t) := \frac{d\boldsymbol{z}_i^{\text{fwd}}(t)}{dt} = g(\boldsymbol{z}_1^{\text{fwd}}(t), \boldsymbol{z}_2^{\text{fwd}}(t), \cdots \boldsymbol{z}_N^{\text{fwd}}(t)),$$
$$\text{where} \quad \boldsymbol{z}_0^{\text{fwd}}(t) = f_{\text{ENC}}(X(t_{1:K}), \mathcal{G}), \quad \hat{\boldsymbol{y}}_i^{\text{fwd}}(t) = f_{\text{DEC}}(\boldsymbol{z}_i^{\text{fwd}}(t)) \tag{6}$$

To train such a GraphODE from data, we can use the reconstruction loss that minimizes the L2 distance between predicted forward trajectories $\hat{\boldsymbol{y}}_i^{\text{fwd}}(t)$ and the ground truth trajectories $\boldsymbol{y}_i(t)$.

$$\mathcal{L}_{pred} = \sum_{i=1}^{N} \sum_{t=0}^{T} \left\| \boldsymbol{y}_i(t) - \hat{\boldsymbol{y}}_i^{\text{fwd}}(t) \right\|_2^2 \tag{7}$$

**Reversed Trajectory Prediction and Regularization Loss.** We now design a novel time-reversal symmetry loss as a soft constraint to flexibly regulate systems' total energy to prevent it from changing sharply. Specifically, based on the definition of time-reversal symmetry in Lemma 1, we first compute the backward trajectories $\boldsymbol{z}^{\text{rev}}$ starting from the ending state of the forward one, traversed back over time:

$$\dot{\boldsymbol{z}}_i^{\text{rev}}(t) := \frac{d\boldsymbol{z}_i^{\text{rev}}(t)}{dt} = -g(\boldsymbol{z}_1^{\text{rev}}(t), \boldsymbol{z}_2^{\text{rev}}(t), \cdots \boldsymbol{z}_N^{\text{rev}}(t)),$$
$$\text{where} \quad \boldsymbol{z}_i^{\text{rev}}(0) = \boldsymbol{z}_i^{\text{fwd}}(T), \quad \hat{\boldsymbol{y}}_i^{rev}(t) = f_{\text{DEC}}(\boldsymbol{z}_i^{\text{rev}}(t)). \tag{8}$$

Next, based on Lemma 1, if the system follows *Time-Reversal Symmetry*, the forward and backward trajectories shall be exactly overlap. We thus design the reversal loss by minimizing the L2 distances

between model forward and backward trajectories:

$$\mathcal{L}_{reverse} = \sum_{i=1}^{N} \sum_{t=0}^{T} \left\| \hat{\boldsymbol{y}}_i^{\mathrm{fwd}}(t) - \hat{\boldsymbol{y}}_i^{\mathrm{rev}}(T-t) \right\|_2^2 \tag{9}$$

Finally, we jointly train TANGO as a weighted combination of the two losses:

$$\mathcal{L} = \mathcal{L}_{pred} + \alpha \mathcal{L}_{reverse} = \sum_{i=1}^{N} \sum_{t=0}^{T} \left\| \boldsymbol{y}_i(t) - \hat{\boldsymbol{y}}_i^{\mathrm{fwd}}(t) \right\|_2^2 + \alpha \sum_{i=1}^{N} \sum_{t=0}^{T} \left\| \hat{\boldsymbol{y}}_i^{\mathrm{fwd}}(t) - \hat{\boldsymbol{y}}_i^{\mathrm{rev}}(T-t) \right\|_2^2, \tag{10}$$

where $\alpha$ is a positive coefficient to balance the two losses based on different targeted systems.

### 3.2 Theoretical Analysis of Time-Reversal Symmetry Loss

We next theoretically show that the time-reversal loss also numerically helps to reduce the prediction loss in general, which can be applicable to various systems under classical mechanics regardless of their physical properties. Specifically, we show that it minimizes higher-order Taylor expansion terms during the ODE integration steps.

**Theorem 1.** *Let $\Delta t$ denote the integration step size in an ODE solver and $T$ be the prediction length. The reconstruction loss $\mathcal{L}_{pred}$ defined in Eq 7 is $\mathcal{O}(T^3 \Delta t^2)$. The time-reversal loss $\mathcal{L}_{reverse}$ defined in Eqn 9 is $\mathcal{O}(T^5 \Delta t^4)$.*

We prove Theorem 1 in Appendix A.2. From Theorem 1, we can see two nice properties of our proposed time-reversal loss: 1) Regarding the relationship to $\Delta t$, $\mathcal{L}_{reverse}$ is optimizing a high-order term $\Delta t^4$, which forces the model to predict fine-grained physical properties such as jerk (the derivatives of accelerations). In comparison, the reconstruction loss optimizes $\Delta t^2$, which mainly guides the model to predict the locations/velocities accurately. Therefore, the combined loss enables our model to be more noise-tolerable; 2) Regarding the relationship to $T$, $\mathcal{L}_{reverse}$ is more sensitive to total sequence length ($T^5$), thus it provides more regularization for long-context prediction, a key challenge for dynamic modeling.

Besides the numerical advantage of TANGO, there are two additional remarks of our method.

**Remark 1.** We define the reversal loss $\mathcal{L}_{reverse}$ as the distance between model forward trajectories and backward trajectories. There are other implementation choices. The first is to minimize the distances between model backward trajectories and ground truth trajectories. When both forward and backward trajectories are close to ground-truth, they are implicitly symmetric. The major drawback is that at the early stage of learning when the forward is far away from ground truth, such implicit regularization does not force time-reversal symmetry, but introduce more noise. Another implementation in TRS-ODE (Huh et al., 2020) follows Eqn 5, where a reverse ODE solves from the starting point, but with opposite velocity. We show that our implementation can have a lower maximum error compared to theirs in A.3. We show the empirical comparison of them in Sec. 4.2.

**Remark 2.** The computational time of the reversal loss $\mathcal{L}_{reverse}$ is of the same scale as the forward reconstruction loss $\mathcal{L}_{pred}$. As the computation process of the reversal loss is to first use the ODE solver to generate the reverse trajectories, which has the same computational overhead as computing the forward trajectories, and then compute the L2 distances. The space complexity is only $\mathcal{O}(1)$ as we are using the adjoint methods (Chen et al., 2018) to compute the reverse trajectories.

## 4 Experiments

**Dynamical Systems and Datasets.** We conduct systematic evaluations over three different spring systems (Kipf et al., 2018) and a complex chaotic pendulum system. For spring systems, we consider three system settings: 1) conservative, i.e. no interactions with the environments, we call it *Simple Spring*; 2) non-conservative with frictions, we call it *Damped Spring*; 3) non-conservative with periodic external forces, we call it *Forced Spring*.

In addition to spring systems, we also consider a chaotic triple *Pendulum* system, which contains three connected sticks in a 2D plane. This *Pendulum* is very sensitive towards initial states, such

Table 1: Evaluation results on MSE ($10^{-2}$). Best results are in **bold** numbers and second-best results are in underline numbers.

| Dataset | Simple Spring | Forced Spring | Damped Spring | Pendulum |
|---|---|---|---|---|
| LatentODE | 5.2622 | 5.0277 | 3.3419 | 2.6894 |
| HODEN | 3.0039 | 4.0668 | 8.7950 | 741.2296 |
| TRS-ODEN | 3.6785 | 4.4465 | 1.7595 | 741.4988 |
| TRS-ODEN$_{GNN}$ | 1.4115 | 2.1102 | 0.5951 | 596.0319 |
| LGODE | 1.7429 | 1.8929 | 0.9718 | 1.4156 |
| TANGO | **1.1178** | **1.4525** | **0.5944** | **1.2527** |
| (—-Ablation of our method with different implementation of $L_{reverse}$—-) | | | | |
| TANGO$_{\mathcal{L}_{rev}=\text{gt-rev}}$ | 1.1313 | 1.5254 | 0.6171 | 1.6158 |
| TANGO$_{\mathcal{L}_{rev}=\text{rev2}}$ | 1.6786 | 1.9786 | 0.9692 | 1.5631 |

that a slight disturbance on the initial states can result in very different trajectories (Shinbrot et al., 1992; Stachowiak & Okada, 2006; Awrejcewicz et al., 2008). Its chaotic nature also results in highly nonlinear and complex trajectories.

Towards physical properties, *Simple Spring* and *Pendulum* are conservative and reversible; *Force Spring* is reversible but non-conservative; *Damped Spring* is irreversible and non-conservative. Details of the datasets and generation pipeline can be found in Appendix C.

**Task Setup.** We evaluate model performance by splitting the trajectories into two halves: $[t_1, t_K]$, $[t_{K+1}, t_T]$ where timestamps can be irregular. We condition the first half of observations to make predictions for the second half as in Rubanova et al. (2019). We generate irregular-sampled trajectories for all datasets and set the number of training samples to be 20,000 and testing samples to be 5,000 respectively. 10% of training samples are used as validation sets in our experiments. The maximum trajectory prediction length is 60.

**Baselines.** We compare TANGO against three baseline types: 1) pure data-driven approaches including LG-ODE (Huang et al., 2020) and LatentODE (Rubanova et al., 2019), where the first one is a multi-agent approach considering the interaction graph, and the second one is a single-agent approach that predicts each trajectory independently; 2) energy-preserving HODEN (Greydanus et al., 2019); and 3) time-reversal TRS-ODEN (Huh et al., 2020).

The latter two are single-agent approaches and require initial states as given input. To handle missing initial states in our dataset, we approximate the initial states for the two methods via linear spline interpolation (Endre Süli, 2003). In addition, we substitute the ODE network in TRS-ODEN with a GNN (Kipf et al., 2018) as TRS-ODEN$_{GNN}$, which serves as a new multi-agent approach, for fair comparison. HODEN cannot be easily extended to the multi-agent setting as replacing the ODE function with a GNN can violate the energy conservation property of the original HODEN. Implementation details of all models can be found in Appendix D.2.

Finally, we conduct two ablation by changing the implementation of $\mathcal{L}_{reverse}$: 1) TANGO$_{\mathcal{L}_{rev}=\text{gt-rev}}$, which computes the reversal loss as the L2 distance between ground truth trajectories to model backward trajectories as discussed in Remark 1 of Sec. 3.2; 2) TANGO$_{\mathcal{L}_{rev}=\text{rev2}}$, which implements the reversal loss based on Eqn 5, similar as TRS-ODEN but calculate over latent $z$. All implementation details is elaborated in Appendix D.2.

## 4.1 MAIN RESULTS

Table 1 presents prediction performance across models measured by mean squared error (MSE). TANGO consistently surpasses other models across datasets, highlighting its generalizability and the efficacy of its reversal loss. Specifically, it reports an improvement in MSE ranging from 11.5% to 34.6% over the second-best baseline. We observe that multi-agent approaches (LG-ODE, TRS-ODEN$_{GNN}$, TANGO) consistently outperform single-agent baselines (LatentODE, HODEN, TRS-ODEN), showing the importance of capturing inter-agent interactions via the message passing of Graph Neural Networks (GNN) encoding and ODE operations.

Table 2: Results of varying observation ratios on MSE ($10^{-2}$).

| Dataset | *Simple Spring* | | *Forced Spring* | | *Damped Spring* | | *Pendulum* | |
|---|---|---|---|---|---|---|---|---|
| Observation Ratios | 0.8 | 0.4 | 0.8 | 0.4 | 0.8 | 0.4 | 0.8 | 0.4 |
| LG-ODE | 1.7054 | 1.6889 | 1.7554 | 2.0370 | 0.9305 | 1.0217 | 1.4314 | 1.7469 |
| TANGO | **1.1176** | **1.1429** | **1.3611** | **1.5109** | **0.6920** | **0.6964** | **1.2309** | **1.2110** |

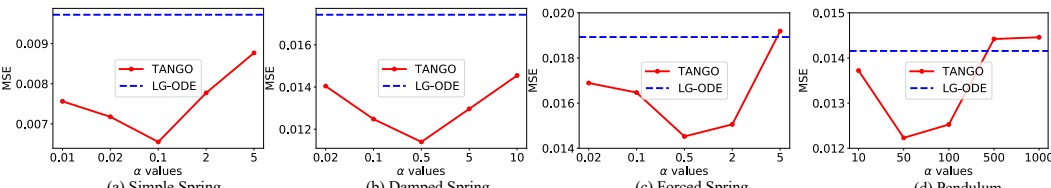

Figure 3: Varying $\alpha$ values across datasets.

When examining individual datasets, HODEN excels among single-agent models on *Simple Spring*, indicating the benefit of incorporating appropriate physical bias. However, its performance is notably lower on *Force Spring* and *Damped Spring*. This suggests that wrongly applied inductive biases can degrade performance. Consequently, while HODEN is suited for strictly energy-conservative systems, TANGO offers versatility across diverse classical dynamics. Note that HODEN naively forces each agent to be energy-conservative, instead of the whole system. Therefore, it still performs worse than multi-agent baselines on energy-conservative systems.

The chaotic nature of the *Pendulum* system, with its sensitivity to initial states [5], poses challenges for dynamic modeling. This lead to pretty unstable predictions for models like HODEN and TRS-ODEN, as these methods rely on linear spline interpolation (Endre Süli, 2003) to approximate missing initial states of agents, which can cause dramatic prediction errors. In contrast, latent models like LatentODE, LG-ODE, and TANGO that utilize advanced encoders derive latent states from observed data, yielding superior accuracy. TANGO maintains the most accurate predictions on such complex *Pendulum* system, further showing its robust generalization capabilities. Finally, we conduct experiments on a real-world motion capture dataset in Appendix E.1.

## 4.2 ABLATION AND SENSITIVITY ANALYSIS

**Ablation on implementation of $\mathcal{L}_{reverse}$.** From the last block of Table 1, we can clearly see that our implementation of reversal loss $\mathcal{L}_{reverse}$ achieves the best performance compared with gt-rev and TRS-ODEN's implementation (rev2). We also analytically show that our reversal loss implementation is expected to achieve a smaller error than the one in TRS-ODEN in Appendix A.3.

**Evaluation across prediction lengths.** We vary the maximum prediction lengths from 20 to 60 across models as shown in Figure 4. As the number of prediction step increases, TANGO keeps the best prediction performance, while other baselines have significant error accumulations. Notably for the chaotic pendulum dataset, which is highly nonlinear and has much more complex inter-agent interactions, single-agent baselines (HODEN, TRS-ODEN) perform poorly even on short prediction lengths. The performance gap between TANGO and baselines increases when making long-range predictions, showing the superior ability of TANGO.

**Evaluation across observation ratios.** For LG-ODE and TANGO, the encoder computes the initial states from observed trajectories. We randomly masked out $40\%$ and $80\%$ observations to study models' sensitivity towards data sparsity. As shown in Table 2, when changing the ratios from $80\%$ to $40\%$, we observe that TANGO has a smaller performance drop compared with LG-ODE, especially on the more complex Pendulum dataset (LG-ODE decreases $22.04\%$ while TANGO decreases $1.62\%$). This indicates that TANGO is less sensitive toward data sparsity.

**Evaluation across different $\alpha$.** We then vary the values of $\alpha$ defined in Eqn 10, which balances the prediction loss and the reversal loss. While prediction loss aims to match true trajectories, reversal

---

[5]Visualization to show *Pendulum* is highly sensitive to different initial states can be found here

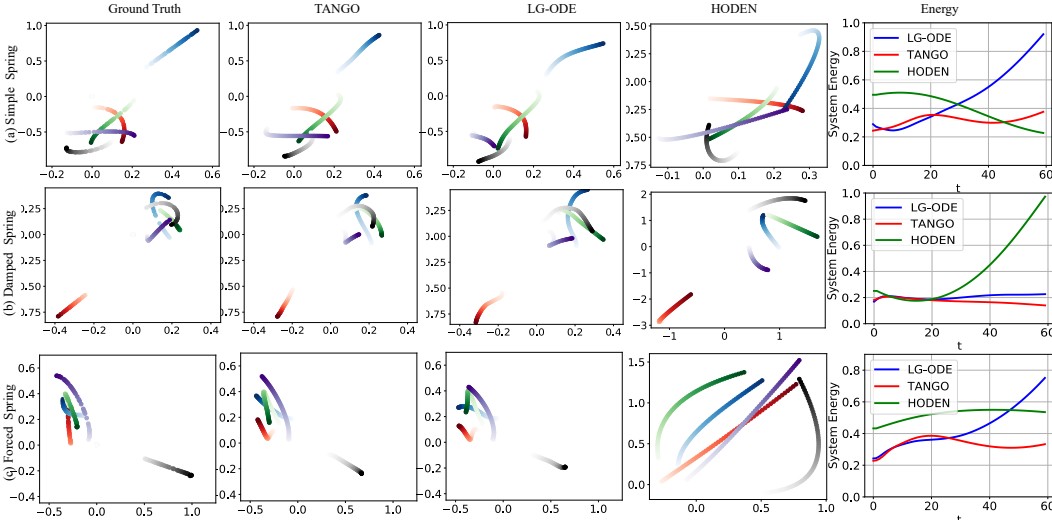

Figure 4: Varying prediction lengths across datasets (Pendulum MSE is in log values.).

Figure 5: Trajectory and energy visualization (trajectory starts from light colors to dark colors.)

loss ensures time-reversal property with better numerical accuracy Figure 3 demonstrates optimal $\alpha$ values being neither too high nor too low. When the weight $\alpha$ is too small, the model tends to neglect the physical bias, reducing test performance; Conversely, very high weights can emphasize reversibility at the cost of accuracy Nonetheless, across various $\alpha$ values, TANGO consistently surpasses LG-ODE, showcasing its flexibility in modeling diverse dynamical systems.

### 4.3 VISUALIZATIONS

**Trajectory Visualizations.** Model predictions and ground truth are visualized in Figure 5. As HODEN is a single-agent baseline that individually forces every agent's energy to be constant over time which is not valid, the predicted trajectories is having the largest errors and systems' total energy is not conserved for all datasets. The purely data-driven LG-ODE exhibits unrealistic energy patterns, as seen in the energy spikes in *Simple Spring* and *Force Spring*. In contrast, TANGO, incorporating reversal loss, generates realistic energy trends, and consistently produces trajectories closest to the ground truth, showing its superior performance. We also visualize the reversal loss over training epochs in Appendix E.3.

## 5 CONCLUSIONS

We propose TANGO, a time-reversible latent GraphODE for modeling multi-agent dynamical systems. TANGO follows the GraphODE framework, with a novel regularization term to softly enforce time-reversal symmetry in the learned systems. Notably, TANGO does not necessitate systems to strictly adhere to energy conservation or reversibility. We unveiled the theoretical reason of this flexibility from a numerical aspect, revealing that the time-reversal loss intrinsically minimizes higher-order terms in the Taylor expansion during the ODE integration. This reduces susceptibility to noise and improves the model's robustness. Empirical evaluations on simulated datasets illustrate TANGO's superior efficacy in capturing real-world system dynamics.

**Ethics Statement.** TANGO is trained upon physical simulation data (e.g., , spring and pendulum) and implemented by public libraries in PyTorch. During the modeling, we neither introduces any social/ethical bias nor amplify any bias in the data. We do not foresee any direct social consequences or ethical issues.

**Reproducibility.** To reproduce our model's results, we provide our code implementation link here. Dataset details can be found in Appendix C and we also provide simulator codes for public use. We also show the implemenmtation details of TANGO and baselines in Apendix D.2.

**Limitations and Future Work.** Currently TANGO only incorporates inductive bias from the temporal aspect, while there are still many important properties in the spatial aspect such as translation and rotation equivariance Satorras et al. (2021). Future endeavors that combine biases from both temporal and spatial dimensions could unveil a new frontier in dynamical systems modeling.

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

# A  THEORETICAL ANALYSIS

## A.1  PROOF OF LEMMA 1

*Proof.* The definition of time-reversal symmetry is given by:

$$R \circ \phi_t = \phi_{-t} \circ R = \phi_t^{-1} \circ R \tag{11}$$

Here, $R$ is an involution operator, which means $R \circ R = \text{id}$.

First, we apply the time evolution operator $\phi_t$ to both sides of Eqn. equation 11:

$$\phi_t \circ R \circ \phi_t = \phi_t \circ \phi_t^{-1} \circ R \tag{12}$$

Simplifying, we obtain:

$$\phi_t \circ R \circ \phi_t = R \tag{13}$$

Next, we apply the involution operator $R$ to both sides of the equation:

$$R \circ \phi_t \circ R \circ \phi_t = R \circ R \tag{14}$$

Since $R \circ R = \text{I}$, we finally arrive at:

$$R \circ \phi_t \circ R \circ \phi_t = \text{I} \tag{15}$$

which means the trajectories can overlap when evolving backward from the final state. $\quad\square$

## A.2  PROOF OF THEOREM 1

Let $\Delta t$ denote the integration step size in an ODE solver and $T$ be the prediction length. The time stamps of the ODE solver are $\{t_j\}_{j=0}^T$, where $t_{j+1} - t_j = \Delta t$ for $j = 0, \cdots, T(T > 1)$. Next suppose during the forward evolution, the updates go through states $\boldsymbol{z}^{\text{fwd}}(t_j) = (\boldsymbol{q}^{\text{fwd}}(t_j), \boldsymbol{p}^{\text{fwd}}(t_j))$ for $j = 0, \cdots, T$, where $\boldsymbol{q}^{\text{fwd}}(t_j)$ is position, $\boldsymbol{p}^{\text{fwd}}(t_j)$ is momentum, while during the reverse evolution they go through states $\boldsymbol{z}^{\text{rev}}(t_j) = (\boldsymbol{q}^{\text{rev}}(t_j), \boldsymbol{p}^{\text{rev}}(t_j))$ for $j = 0, \cdots, T$, in reverse order. The ground truth trajectory is $\boldsymbol{z}^{\text{gt}}(t_j) = (\boldsymbol{q}^{\text{gt}}(t_j), \boldsymbol{p}^{\text{gt}}(t_j))$ for $j = 0, \cdots, T$.

For the sake of brevity in the ensuing proof, we denote $\boldsymbol{z}^{\text{gt}}(t_j)$ by $\boldsymbol{z}_j^{\text{gt}}$, $\boldsymbol{z}^{\text{fwd}}(t_j)$ by $\boldsymbol{z}_j^{\text{fwd}}$ and $\boldsymbol{z}^{\text{rev}}(t_j)$ by $\boldsymbol{z}_j^{\text{rev}}$, and we will use Mathematical Induction to prove the theorem.

### A.2.1  RECONSTRUCTION LOSS ($\mathcal{L}_{pred}$) ANALYSIS.

First, we bound the forward loss $\sum_{j=0}^T \|\boldsymbol{z}_j^{\text{fwd}} - \boldsymbol{z}_j^{\text{gt}}\|_2^2$. Since our method models the momentum and position of the system, we can write the following Taylor expansion of the forward process, where for any $0 \leq j < T$:

$$\begin{cases} \boldsymbol{q}_{j+1}^{\text{fwd}} = \boldsymbol{q}_j^{\text{fwd}} + (\boldsymbol{p}_j^{\text{fwd}}/m)\Delta t + (\dot{\boldsymbol{p}}_j^{\text{fwd}}/2m)\Delta t^2 + \mathcal{O}(\Delta t^3), & \text{(16a)} \\ \boldsymbol{p}_{j+1}^{\text{fwd}} = \boldsymbol{p}_j^{\text{fwd}} + \dot{\boldsymbol{p}}_j^{\text{fwd}}\Delta t + \mathcal{O}(\Delta t^2), & \text{(16b)} \\ \dot{\boldsymbol{p}}_{j+1}^{\text{fwd}} = \dot{\boldsymbol{p}}_j^{\text{fwd}} + \mathcal{O}(\Delta t), & \text{(16c)} \end{cases}$$

and for the ground truth process, we also have from Taylor expansion that

$$\begin{cases} \boldsymbol{q}_{j+1}^{\text{gt}} = \boldsymbol{q}_j^{\text{gt}} + (\boldsymbol{p}_j^{\text{gt}}/m)\Delta t + (\dot{\boldsymbol{p}}_j^{\text{gt}}/2m)\Delta t^2 + \mathcal{O}(\Delta t^3), & \text{(17a)} \\ \boldsymbol{p}_{j+1}^{\text{gt}} = \boldsymbol{p}_j^{\text{gt}} + \dot{\boldsymbol{p}}_j^{\text{gt}}\Delta t + \mathcal{O}(\Delta t^2), & \text{(17b)} \\ \dot{\boldsymbol{p}}_{j+1}^{\text{gt}} = \dot{\boldsymbol{p}}_j^{\text{gt}} + \mathcal{O}(\Delta t). & \text{(17c)} \end{cases}$$

With these, we aim to prove that for any $k = 0, 1, \cdots, T$, the following hold :

$$\begin{cases} \|\boldsymbol{q}_k^{\text{fwd}} - \boldsymbol{q}_k^{\text{gt}}\|_2 \leq C_2^{\text{fwd}} k^2 \Delta t^2, & \text{(18a)} \\ \|\boldsymbol{p}_k^{\text{fwd}} - \boldsymbol{p}_k^{\text{gt}}\|_2 \leq C_1^{\text{fwd}} k \Delta t, & \text{(18b)} \end{cases}$$

where $C_1^{\text{fwd}}$ and $C_2^{\text{fwd}}$ are constants.

**Base Case** $k = 0$: Based on the initialization rules, it is obvious that $\left\|\boldsymbol{q}_0^{\text{fwd}} - \boldsymbol{q}_0^{\text{gt}}\right\|_2 = 0$ and $\left\|\boldsymbol{p}_0^{\text{fwd}} - \boldsymbol{p}_0^{\text{gt}}\right\|_2 = 0$, thus (18a) and (18b) both hold for $k = 0$.

**Inductive Hypothesis:** Assume (18a) and (18b) hold for $k = j$, which means:

$$\begin{cases} \|\boldsymbol{q}_j^{\text{fwd}} - \boldsymbol{q}_j^{\text{gt}}\|_2 \leq C_2^{\text{fwd}} j^2 \Delta t^2, & \text{(19a)} \\ \|\boldsymbol{p}_j^{\text{fwd}} - \boldsymbol{p}_j^{\text{gt}}\|_2 \leq C_1^{\text{fwd}} j \Delta t, & \text{(19b)} \end{cases}$$

**Inductive Proof:** We need to prove (18a) and (18b) hold for $k = j + 1$.

First, using (16c) and (17c), we have

$$\left\|\dot{\boldsymbol{p}}_{j+1}^{\text{fwd}} - \dot{\boldsymbol{p}}_{j+1}^{\text{gt}}\right\|_2 = \left\|\dot{\boldsymbol{p}}_j^{\text{fwd}} - \dot{\boldsymbol{p}}_j^{\text{gt}}\right\|_2 + \mathcal{O}(\Delta t) = \left\|\dot{\boldsymbol{p}}_0^{\text{fwd}} - \dot{\boldsymbol{p}}_0^{\text{gt}}\right\|_2 + \mathcal{O}\big((j+1)\Delta t\big) = \mathcal{O}(1), \quad (20)$$

where we iterate through $j, j-1, \cdots, 0$ in the second equality. Then using (17b) and (16b), we get for $j + 1$ that

$$\begin{aligned} \left\|\boldsymbol{p}_{j+1}^{\text{fwd}} - \boldsymbol{p}_{j+1}^{\text{gt}}\right\|_2 &= \left\|\big(\boldsymbol{p}_j^{\text{fwd}} + \dot{\boldsymbol{p}}_j^{\text{fwd}}\Delta t\big) - \big(\boldsymbol{p}_j^{\text{gt}} + \dot{\boldsymbol{p}}_j^{\text{gt}}\Delta t\big) + \mathcal{O}(\Delta t^2)\right\|_2 \\ &\leq \left\|\boldsymbol{p}_j^{\text{fwd}} - \boldsymbol{p}_j^{\text{gt}}\right\|_2 + \left\|\dot{\boldsymbol{p}}_j^{\text{fwd}} - \dot{\boldsymbol{p}}_j^{\text{gt}}\right\|_2 \Delta t + \mathcal{O}(\Delta t^2) \\ &\leq \left[C_1^{\text{fwd}} j + \mathcal{O}(1)\right]\Delta t, \end{aligned}$$

where the first inequality uses the triangle inequality, and in the second inequality we use (19b) as well as (20). We can see there exists $C_1^{\text{fwd}}$ such that the final expression above is upper bounded by $C_1^{\text{fwd}}(j+1)\Delta t$, with which the claim holds for $j + 1$.

Next for (18a), using (17a) and (16a), we get for any $j$ that

$$\begin{aligned} \left\|\boldsymbol{q}_{j+1}^{\text{fwd}} - \boldsymbol{q}_{j+1}^{\text{gt}}\right\|_2 &= \big\|\big(\boldsymbol{q}_j^{\text{fwd}} + (\boldsymbol{p}_j^{\text{fwd}}/m)\Delta t + (\dot{\boldsymbol{p}}_j^{\text{fwd}}/2m)\Delta t^2\big) \\ &\quad - \big(\boldsymbol{q}_j^{\text{gt}} + (\boldsymbol{p}_j^{\text{gt}}/m)\Delta t + (\dot{\boldsymbol{p}}_j^{\text{gt}}/2m)\Delta t^2\big) + \mathcal{O}(\Delta t^3)\big\|_2 \\ &\leq \left\|\boldsymbol{q}_j^{\text{fwd}} - \boldsymbol{q}_j^{\text{gt}}\right\|_2 + \frac{1}{m}\left\|\boldsymbol{p}_j^{\text{fwd}} - \boldsymbol{p}_j^{\text{gt}}\right\|_2 \Delta t + \frac{1}{2m}\left\|\dot{\boldsymbol{p}}_j^{\text{fwd}} - \dot{\boldsymbol{p}}_j^{\text{gt}}\right\|_2 \Delta t^2 + \mathcal{O}(\Delta t^3) \\ &\leq \left[C_2^{\text{fwd}} j^2 + \frac{C_1^{\text{fwd}}}{m} j + \mathcal{O}(1)\right]\Delta t^2, \end{aligned}$$

where the first inequality uses the triangle inequality, and in the second inequality we use (19a) and (19b) as well as (20). Thus with an appropriate $C_2^{\text{fwd}}$, we have the final expression above is upper bounded by $C_2^{\text{fwd}}(j+1)^2 \Delta t^2$, and so the claim holds for $j + 1$.

Since both the base case and the inductive step have been proven, by the principle of mathematical induction, (18a) and (18b) holds for all $k = 0, 1, \cdots, T$.

With this, we finish the forward proof by plugging (18a) and (18b) into the loss function:

$$\begin{aligned} \sum_{j=0}^{T} \|\boldsymbol{z}_j^{\text{fwd}} - \boldsymbol{z}_j^{\text{gt}}\|_2^2 &= \sum_{j=0}^{T} \|\boldsymbol{p}_j^{\text{fwd}} - \boldsymbol{p}_j^{\text{gt}}\|_2^2 + \sum_{j=0}^{T} \|\boldsymbol{q}_j^{\text{fwd}} - \boldsymbol{q}_j^{\text{gt}}\|_2^2 \\ &\leq \big(C_1^{\text{fwd}}\big)^2 \sum_{j=0}^{T} j^2 \Delta t^2 + \big(C_2^{\text{fwd}}\big)^2 \sum_{j=0}^{T} j^4 \Delta t^4 \\ &= \mathcal{O}(T^3 \Delta t^2). \end{aligned}$$

### A.2.2 REVERSAL LOSS ($\mathcal{L}_{reverse}$) ANALYSIS.

Next we analyze the reversal loss $\sum_{j=0}^{T} \|R(\boldsymbol{z}_j^{\text{rev}}) - \boldsymbol{z}_j^{\text{fwd}}\|_2^2$. For this, we need to refine the Taylor expansion residual terms for a more in-depth analysis.

First reconsider the forward process. Since the process is generated from the learned network, we may assume that for some constants $c_1$, $c_2$, and $c_3$, the states satisfy the following for any $0 \leq j < T$:

$$\begin{cases} \boldsymbol{q}_j^{\text{fwd}} = \boldsymbol{q}_{j+1}^{\text{fwd}} - (\boldsymbol{p}_{j+1}^{\text{fwd}}/m)\Delta t + (\dot{\boldsymbol{p}}_{j+1}^{\text{fwd}}/2m)\Delta t^2 + \mathbf{rem}_j^{\text{fwd},3}, & \text{(21a)} \\ \boldsymbol{p}_j^{\text{fwd}} = \boldsymbol{p}_{j+1}^{\text{fwd}} - \dot{\boldsymbol{p}}_{j+1}^{\text{fwd}}\Delta t + \mathbf{rem}_j^{\text{fwd},2}, & \text{(21b)} \\ \dot{\boldsymbol{p}}_j^{\text{fwd}} = \dot{\boldsymbol{p}}_{j+1}^{\text{fwd}} + \mathbf{rem}_j^{\text{fwd},1}, & \text{(21c)} \end{cases}$$

where the remaining terms $\left\|\mathbf{rem}_j^{\mathrm{fwd},i}\right\|_2 \leq c_i \Delta t^i$ for $i = 1, 2, 3$. Similarly, we have approximate Taylor expansions for the reverse process:

$$\begin{cases} \boldsymbol{q}_j^{\mathrm{rev}} = \boldsymbol{q}_{j+1}^{\mathrm{rev}} + (\boldsymbol{p}_{j+1}^{\mathrm{rev}}/m)\Delta t + (\dot{\boldsymbol{p}}_{j+1}^{\mathrm{rev}}/2m)\Delta t^2 + \mathbf{rem}_j^{\mathrm{rev},3}, & \text{(22a)} \\[4pt] \boldsymbol{p}_j^{\mathrm{rev}} = \boldsymbol{p}_{j+1}^{\mathrm{rev}} + \dot{\boldsymbol{p}}_{j+1}^{\mathrm{rev}}\Delta t + \mathbf{rem}_j^{\mathrm{rev},2}, & \text{(22b)} \\[4pt] \dot{\boldsymbol{p}}_j^{\mathrm{rev}} = \dot{\boldsymbol{p}}_{j+1}^{\mathrm{rev}} + \mathbf{rem}_j^{\mathrm{rev},1}, & \text{(22c)} \end{cases}$$

where $\left\|\mathbf{rem}_j^{\mathrm{rev},i}\right\|_2 \leq c_i \Delta t^i$ for $i = 1, 2, 3$.

We will prove via induction that for $k = T, T-1, \cdots, 0$,

$$\begin{cases} \|R(\boldsymbol{q}_k^{\mathrm{rev}}) - \boldsymbol{q}_k^{\mathrm{fwd}}\|_2 \leq C_3^{\mathrm{rev}}(T-k)^3\Delta t^3, & \text{(23a)} \\[4pt] \|R(\boldsymbol{p}_k^{\mathrm{rev}}) - \boldsymbol{p}_k^{\mathrm{fwd}}\|_2 \leq C_2^{\mathrm{rev}}(T-k)^2\Delta t^2, & \text{(23b)} \\[4pt] \|R(\dot{\boldsymbol{p}}_k^{\mathrm{rev}}) - \dot{\boldsymbol{p}}_k^{\mathrm{fwd}}\|_2 \leq C_1^{\mathrm{rev}}(T-k)\Delta t, & \text{(23c)} \end{cases}$$

where $C_1^{\mathrm{rev}}$, $C_2^{\mathrm{rev}}$ and $C_3^{\mathrm{rev}}$ are constants.

The entire proof process is analogous to the previous analysis of Reconstruction Loss.

**Base Case $k = T$:** Since the reverse process is initialized by the forward process variables at $k = T$, it is obvious that $\left\|\boldsymbol{q}_T^{\mathrm{fwd}} - \boldsymbol{q}_T^{ev}\right\|_2 = \left\|\boldsymbol{p}_T^{\mathrm{fwd}} - \boldsymbol{p}_T^{\mathrm{rev}}\right\|_2 = \left\|\dot{\boldsymbol{p}}_T^{\mathrm{fwd}} - \dot{\boldsymbol{p}}_T^{\mathrm{rev}}\right\|_2 = 0$. Thus (23a), (23b) and (23c) all hold for $k = 0$.

**Inductive Hypothesis:** Assume the inequalities (23b), (23a) and (23c) hold for $k = j + 1$, which means:

$$\begin{cases} \|R(\boldsymbol{q}_{j+1}^{\mathrm{rev}}) - \boldsymbol{q}_{j+1}^{\mathrm{fwd}}\|_2 \leq C_3^{\mathrm{rev}}(T-(j+1))^3\Delta t^3, & \text{(24a)} \\[4pt] \|R(\boldsymbol{p}_{j+1}^{\mathrm{rev}}) - \boldsymbol{p}_{j+1}^{\mathrm{fwd}}\|_2 \leq C_2^{\mathrm{rev}}(T-(j+1))^2\Delta t^2, & \text{(24b)} \\[4pt] \|R(\dot{\boldsymbol{p}}_{j+1}^{\mathrm{rev}}) - \dot{\boldsymbol{p}}_{j+1}^{\mathrm{fwd}}\|_2 \leq C_1^{\mathrm{rev}}(T-(j+1))\Delta t, & \text{(24c)} \end{cases}$$

**Inductive Proof:** We need to prove (23b) (23a) and (23c) holds for $k = j$.

First, for (23c), using (21c) and (22c), we get for any $j$ that

$$\begin{aligned} \left\|R(\dot{\boldsymbol{p}}_j^{\mathrm{rev}}) - \dot{\boldsymbol{p}}_j^{\mathrm{fwd}}\right\|_2 &= \left\|(\dot{\boldsymbol{p}}_{j+1}^{\mathrm{rev}} + \mathbf{rem}_j^{\mathrm{rev},1}) - (\dot{\boldsymbol{p}}_{j+1}^{\mathrm{fwd}} + \mathbf{rem}_j^{\mathrm{fwd},1})\right\|_2 \\ &\leq \left\|R(\dot{\boldsymbol{p}}_{j+1}^{\mathrm{rev}}) - \dot{\boldsymbol{p}}_{j+1}^{\mathrm{fwd}}\right\|_2 + \left\|\mathbf{rem}_j^{\mathrm{rev},1}\right\|_2 + \left\|\mathbf{rem}_j^{\mathrm{fwd},1}\right\|_2 \\ &\leq C_1^{\mathrm{rev}}(T-j-1)\Delta t + 2c_1\Delta t, \end{aligned}$$

where the first inequality uses the triangle inequality, and the second inequality plugs in (24c). Thus taking $C_1^{\mathrm{rev}} = 2c_1$, the above is upped bounded by $C_1^{\mathrm{rev}}(T-j)\Delta t$, and (23b) holds for $j$.

Second, for (24b), using (21b) and (22b), we get

$$\begin{aligned} \left\|R(\boldsymbol{p}_j^{\mathrm{rev}}) - \boldsymbol{p}_j^{\mathrm{fwd}}\right\|_2 &= \left\| - \left(\boldsymbol{p}_{j+1}^{\mathrm{rev}} + \dot{\boldsymbol{p}}_{j+1}^{\mathrm{rev}}\Delta t + \mathbf{rem}_j^{\mathrm{rev},2}\right) - \left(\boldsymbol{p}_{j+1}^{\mathrm{fwd}} - \dot{\boldsymbol{p}}_{j+1}^{\mathrm{fwd}}\Delta t + \mathbf{rem}_j^{\mathrm{fwd},2}\right)\right\|_2 \\ &\leq \left\|R(\boldsymbol{p}_{j+1}^{\mathrm{rev}}) - \boldsymbol{p}_{j+1}^{\mathrm{fwd}}\right\|_2 + \left\|R(\dot{\boldsymbol{p}}_{j+1}^{\mathrm{rev}}) - \dot{\boldsymbol{p}}_{j+1}^{\mathrm{fwd}}\right\|_2 \Delta t \\ &\quad + \left\|\mathbf{rem}_j^{\mathrm{rev},2}\right\|_2 + \left\|\mathbf{rem}_j^{\mathrm{fwd},2}\right\|_2 \\ &\leq \left[C_2^{\mathrm{rev}}(T-j-1)^2 + C_1^{\mathrm{rev}}(T-j-1) + 2c_2\right]\Delta t^2, \end{aligned}$$

where the first inequality uses the triangle inequality, and in the second inequality we use (24a) and (24b). Thus taking $C_2^{\mathrm{rev}} = \max\{C_1^{\mathrm{rev}}/2, 2c_2\}$, we have the final expression above is upper bounded by $C_2^{\mathrm{rev}}(T-j)^2\Delta t^2$, and so the claim holds for $j$.

Finally, for (24a), we use (21a) and (22a) to get

$$\begin{aligned} \left\|R(\boldsymbol{q}_j^{\mathrm{rev}}) - \boldsymbol{q}_j^{\mathrm{fwd}}\right\|_2 &= \left\|\left(\boldsymbol{q}_{j+1}^{\mathrm{rev}} + (\boldsymbol{p}_{j+1}^{\mathrm{rev}}/m)\Delta t + (\dot{\boldsymbol{p}}_{j+1}^{\mathrm{rev}}/2m)\Delta t^2 + \mathbf{rem}_j^{\mathrm{rev},3}\right) \right.\\ &\quad \left. - \left(\boldsymbol{q}_{j+1}^{\mathrm{fwd}} - (\boldsymbol{p}_{j+1}^{\mathrm{fwd}}/m)\Delta t + (\dot{\boldsymbol{p}}_{j+1}^{\mathrm{fwd}}/2m)\Delta t^2 + \mathbf{rem}_j^{\mathrm{fwd},3}\right)\right\|_2 \\ &\leq \left\|R(\boldsymbol{q}_{j+1}^{\mathrm{rev}}) - \boldsymbol{q}_{j+1}^{\mathrm{fwd}}\right\|_2 + \frac{1}{m}\left\|R(\boldsymbol{p}_{j+1}^{\mathrm{rev}}) - \boldsymbol{p}_{j+1}^{\mathrm{fwd}}\right\|_2\Delta t \\ &\quad + \frac{1}{2m}\left\|R(\dot{\boldsymbol{p}}_{j+1}^{\mathrm{rev}}) - \dot{\boldsymbol{p}}_{j+1}^{\mathrm{fwd}}\right\|_2\Delta t^2 + \left\|\mathbf{rem}_j^{\mathrm{rev},3}\right\|_2 + \left\|\mathbf{rem}_j^{\mathrm{fwd},3}\right\|_2 \\ &\leq \left[C_3^{\mathrm{rev}}(T-j-1)^3 + \frac{C_2^{\mathrm{rev}}}{m}(T-j-1)^2 + \frac{C_1^{\mathrm{rev}}}{2m}(T-j-1) + 2c_3\right]\Delta t^3, \end{aligned}$$

where the first inequality uses the triangle inequality, and in the second inequality we use (24a), (24b) and (24c). Thus taking $C_3^{\text{rev}} = \max\{C_2^{\text{rev}}/3m, C_1^{\text{rev}}/6m, 2c_3\}$, we have the final expression above is upper bounded by $C_3^{\text{rev}}(T-j)^3\Delta t^3$, and so the claim holds for $j$.

Since both the base case and the inductive step have been proven, by the principle of mathematical induction, (23b), (23a) and (23c) hold for all $k = T, T-1, \cdots, 0$.

With this we finish the proof by plugging (23b) and (23a) into the loss function:

$$\sum_{j=0}^{T}\|R(\boldsymbol{z}_j^{\text{rev}}) - \boldsymbol{z}_j^{\text{fwd}}\|_2^2 = \sum_{j=0}^{T}\|R(\boldsymbol{p}_j^{\text{rev}}) - \boldsymbol{p}_j^{\text{fwd}}\|_2^2 + \sum_{j=0}^{T}\|R(\boldsymbol{q}_j^{\text{rev}}) - \boldsymbol{q}_j^{\text{fwd}}\|_2^2$$

$$\le \left(C_2^{\text{rev}}\right)^2 \sum_{j=0}^{T}(T-j)^4\Delta t^4 + \left(C_3^{\text{rev}}\right)^2 \sum_{j=0}^{T}(T-j)^6\Delta t^6$$

$$= \mathcal{O}(T^5\Delta t^4).$$

### A.3 ANALYSIS ON IMPLEMENTATIONS OF REVERSAL LOSS

Our time-reversal loss implementation builts upon Lemma1 where the backward trajectory originates from the last state of the forward trajectory. One could also implement the reversal loss based on Eqn 5 which is adopted in TRS-ODEN (Huh et al., 2020). We illustrated the comparison between the two implementation in Figure 6.

**Remark 3.** Comparing TANGO against the implementation following Eqn.(5), when the reconstruction loss defined in Eqn.(7) and the time-reversal loss defined in Eqn. (9) of both methods are equal, the maximum error between the reversal and ground truth trajectory, i.e. $MaxError_{gt\_rev} = \max_{j\in[T]}\|\boldsymbol{y}(j) - \hat{\boldsymbol{y}}^{\text{rev}}(T-j)\|_2$ made by TANGO is smaller than that of following Eqn.(5).

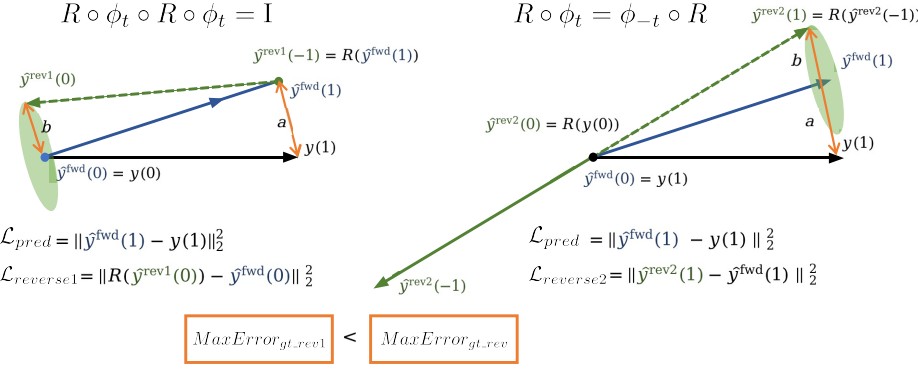

Figure 6: Comparison between two reversal loss implementation

This remark suggests that our implementation of time-reversal symmetry is numerically better than the implementation used in (Huh et al., 2020). We give an illustration of the remark below.

We expect an ideal model to align both the predicted forward and reverse trajectories with the ground truth. As shown in Figure 6, we integrate one step from the initial state $\hat{\boldsymbol{y}}^{\text{fwd}}(0)$ (which is the same as $\boldsymbol{y}(0)$) and reach the state $\hat{\boldsymbol{y}}^{\text{fwd}}(1)$.

The first reverse loss implementation (ours) follows lemma (1) as $R\circ\Phi_t\circ R\circ\Phi_t = id$, which means when we evolve forward and reach the state $\hat{\boldsymbol{y}}^{\text{fwd}}(1)$ we reverse it into $\hat{\boldsymbol{y}}^{\text{rev1}}(-1) = R(\hat{\boldsymbol{y}}^{\text{fwd}}(1))$ and go back to reach $\hat{\boldsymbol{y}}^{\text{rev1}}(0)$, then reverse it to get $R(\hat{\boldsymbol{y}}^{\text{rev1}}(0))$, which ideally should be the same as $\hat{\boldsymbol{y}}^{\text{fwd}}(0)$.

The second reverse loss implementation follows Eqn.(5) as $R\circ\Phi_t = \Phi_{-t}\circ R$, which means we first reverse the initial state as $\hat{\boldsymbol{y}}^{\text{rev2}}(0) = R(\boldsymbol{y}(0))$, then evolve the reverse trajectory in the opposite direction to reach $\hat{\boldsymbol{y}}^{\text{rev2}}(-1)$, and then perform a symmetric operation to reach $\hat{\boldsymbol{y}}^{\text{rev2}}(1)$, aligning it with the forward trajectory.

We assume the two reconstruction losses $\mathcal{L}_{pred} = \|\hat{\boldsymbol{y}}^{\text{fwd}}(1) - \boldsymbol{y}(1)\|_2^2 := a$ are the same. For the time-reversal losses, we also assume they have reached the same value $b$:

$$\mathcal{L}_{reverse1} = \|R(\hat{\boldsymbol{y}}^{\text{rev1}}(0)) - \hat{\boldsymbol{y}}^{\text{fwd}}(0)\|_2^2 + \|R(\hat{\boldsymbol{y}}^{\text{rev1}}(-1)) - \hat{\boldsymbol{y}}^{\text{fwd}}(1)\|_2^2 = \|R(\hat{\boldsymbol{y}}^{\text{rev1}}(0)) - \hat{\boldsymbol{y}}^{\text{fwd}}(0)\|_2^2 := b,$$

$$\mathcal{L}_{reverse2} = \|\hat{\boldsymbol{y}}^{\text{rev2}}(0) - \hat{\boldsymbol{y}}^{\text{fwd}}(0)\|_2^2 + \|\hat{\boldsymbol{y}}^{\text{rev2}}(1) - \hat{\boldsymbol{y}}^{\text{fwd}}(1)\|_2^2 = \|\hat{\boldsymbol{y}}^{\text{rev2}}(1) - \hat{\boldsymbol{y}}^{\text{fwd}}(1)\|_2^2 := b,$$

As shown in Figure.6 where we illustrate the worst case scenario $MaxError_{gt\_rev} = \max_{j \in [T]} \|\boldsymbol{y}(j) - \hat{\boldsymbol{y}}^{\text{rev}}(T-j)\|_2$ of the two loss, we can see that in our implementation the worst error is the maximum of two loss, while the TRS-ODEN's implementation has the risk of accumulating the error together, making the worst error being the sum of both:

$$MaxError_{gt\_rev1} = \max\left\{\|R(\hat{\boldsymbol{y}}^{\text{rev1}}(0)) - \boldsymbol{y}(0)\|_2, \|R(\hat{\boldsymbol{y}}^{\text{rev1}}(-1)) - \boldsymbol{y}(1)\|_2\right\} = max\{a, b\},$$

$$\begin{aligned}
MaxError_{gt\_rev2} &= \max\left\{\|\hat{\boldsymbol{y}}^{\text{rev2}}(0) - \boldsymbol{y}(0)\|_2, \|\hat{\boldsymbol{y}}^{\text{rev2}}(1) - \boldsymbol{y}(1)\|_2\right\} \\
&= \max\left\{0, \|R(\hat{\boldsymbol{y}}^{\text{rev1}}(-1)) - \boldsymbol{y}(1)\|_2\right\} \\
&= \|\hat{\boldsymbol{y}}^{\text{rev2}}(1) - \hat{\boldsymbol{y}}^{\text{fwd}}(1)\|_2 + \|\hat{\boldsymbol{y}}^{\text{fwd}}(1) - \boldsymbol{y}(1)\|_2 = a + b,
\end{aligned}$$

So it is obvious that $MaxError_{gt\_rev2} > MaxError_{gt\_rev1}$, which means our model achieves a smaller error of the maximum distance between the reversal and ground truth trajectory.

## B EXAMPLE OF VARYING DYNAMICAL SYSTEMS

We illustrate the energy conservation and time reversal of the three n-body spring systems in Figure 1(a). We use the Hamiltonian formalism of systems under classical mechanics to describe their dynamics and verify their energy conservation and time-reversibility characteristics.

The scalar function that describes a systems motion is called the Hamiltonian, $\mathcal{H}$, and is typically equal to the total energy of the system, that is, the potential energy plus the kinetic energy (North, 2021). It describes the phase space equations of motion by following two first-order ODEs called Hamilton's equations:

$$\frac{d\mathbf{q}}{dt} = \frac{\partial \mathcal{H}(\mathbf{q}, \mathbf{p})}{\partial \mathbf{p}}, \frac{d\mathbf{p}}{dt} = -\frac{\partial \mathcal{H}(\mathbf{q}, \mathbf{p})}{\partial \mathbf{q}}, \tag{25}$$

where $\mathbf{q} \in \mathbb{R}^n, \mathbf{p} \in \mathbb{R}^n$, and $\mathcal{H} : \mathbb{R}^{2n} \mapsto \mathbb{R}$ are positions, momenta, and Hamiltonian of the system.

Under this formalism, energy conservative is defined by $d\mathcal{H}/dt = 0$, and the time-reversal symmetry is defined by $\mathcal{H}(q, p, t) = \mathcal{H}(q, -p, -t)$ (Lamb & Roberts, 1998).

### B.1 CONSERVATIVE AND REVERSIBLE SYSTEMS.

A simple example is the isolated n-body spring system, which can be described by :

$$\begin{aligned}
\frac{d\mathbf{q_i}}{dt} &= \frac{\mathbf{p_i}}{m} \\
\frac{d\mathbf{p_i}}{dt} &= \sum_{j \in N_i} -k(\mathbf{q_i} - \mathbf{q_j}),
\end{aligned} \tag{26}$$

where $\mathbf{q} = (\mathbf{q_1}, \mathbf{q_2}, \cdots, \mathbf{q_N})$ is a set of positions of each object , $\mathbf{p} = (\mathbf{p_1}, \mathbf{p_2}, \cdots, \mathbf{p_N})$ is a set of momenta of each object, $m_i$ is mass of each object, $k$ is spring constant.

The Hamilton's equations are:

$$\begin{aligned}
\frac{\partial \mathcal{H}(\mathbf{q}, \mathbf{p})}{\partial \mathbf{p_i}} &= \frac{d\mathbf{q_i}}{dt} = \frac{\mathbf{p_i}}{m} \\
\frac{\partial \mathcal{H}(\mathbf{q}, \mathbf{p})}{\partial \mathbf{q_i}} &= -\frac{d\mathbf{p_i}}{dt} = \sum_{j \in N_i} k(\mathbf{q_i} - \mathbf{q_j}),
\end{aligned} \tag{27}$$

Hence, we can obtain the Hamiltonian through the integration of the above equation.

$$\mathcal{H}(\mathbf{q}, \mathbf{p}) = \sum_{i=1}^{N} \frac{\mathbf{p_i}^2}{2m_i} + \frac{1}{2} \sum_{i=1}^{N} \sum_{j \in N_i}^{N} \frac{1}{2} k(\mathbf{q_i} - \mathbf{q_j})^2, \tag{28}$$

**Verify the systems' energy conservation**

$$\frac{d\mathcal{H}(\mathbf{q},\mathbf{p})}{dt} = \frac{1}{dt}\Big(\sum_{i=1}^{N}\frac{\mathbf{p_i}^2}{2m_i}\Big) + \frac{1}{dt}\Big(\frac{1}{2}\sum_{i=1}^{N}\sum_{j\in N_i}^{N}\frac{1}{2}k(\mathbf{q_i}-\mathbf{q_j})^2\Big) = 0, \tag{29}$$

So it is conservative.

**Verify the systems' time-reversal symmetry** We do the transformation $R \; : \; (\mathbf{q},\mathbf{p},t) \mapsto (\mathbf{q},-\mathbf{p},-t)$.

$$\mathcal{H}(\mathbf{q},\mathbf{p}) = \sum_{i=1}^{N}\frac{\mathbf{p_i}^2}{2m_i} + \frac{1}{2}\sum_{i=1}^{N}\sum_{j\in N_i}^{N}\frac{1}{2}k(\mathbf{q_i}-\mathbf{q_j})^2,$$

$$\mathcal{H}(\mathbf{q},-\mathbf{p}) = \sum_{i=1}^{N}\frac{(-\mathbf{p_i})^2}{2m_i} + \frac{1}{2}\sum_{i=1}^{N}\sum_{j\in N_i}^{N}\frac{1}{2}k(\mathbf{q_i}-\mathbf{q_j})^2, \tag{30}$$

It is obvious $\mathcal{H}(\mathbf{q},\mathbf{p}) = \mathcal{H}(\mathbf{q},-\mathbf{p})$, so it is reversible

### B.2 NON-CONSERVATIVE AND REVERSIBLE SYSTEMS.

A simple example is a n-body spring system with periodical external force, which can be described by:

$$\frac{d\mathbf{q_i}}{dt} = \frac{\mathbf{p_i}}{m}$$
$$\frac{d\mathbf{p_i}}{dt} = \sum_{j\in N_i}^{N} -k(\mathbf{q_i}-\mathbf{q_j}) - k_1\cos\omega t, \tag{31}$$

The Hamilton's equations are:

$$\frac{\partial\mathcal{H}(\mathbf{q},\mathbf{p})}{\partial\mathbf{p_i}} = \frac{d\mathbf{q_i}}{dt} = \frac{\mathbf{p_i}}{m}$$
$$\frac{\partial\mathcal{H}(\mathbf{q},\mathbf{p})}{\partial\mathbf{q_i}} = -\frac{d\mathbf{p_i}}{dt} = \sum_{j\in N_i} k(\mathbf{q_i}-\mathbf{q_j}) + k_1\cos\omega t, \tag{32}$$

Hence, we can obtain the Hamiltonian through the integration of the above equation:

$$\mathcal{H}(\mathbf{q},\mathbf{p}) = \sum_{i=1}^{N}\frac{\mathbf{p_i}^2}{2m_i} + \frac{1}{2}\sum_{i=1}^{N}\sum_{j\in N_i}^{N}\frac{1}{2}k(\mathbf{q_i}-\mathbf{q_j})^2 + \sum_{i=1}^{N}q_i * k_1\cos\omega t, \tag{33}$$

**Verify the systems' energy conservation**

$$\begin{aligned}\frac{d\mathcal{H}(\mathbf{q},\mathbf{p})}{dt} &= \frac{1}{dt}\Big(\sum_{i=1}^{N}\frac{\mathbf{p_i}^2}{2m_i}\Big) + \frac{1}{dt}\Big(\frac{1}{2}\sum_{i=1}^{N}\sum_{j\in N_i}^{N}\frac{1}{2}k(\mathbf{q_i}-\mathbf{q_j})^2\Big) + \frac{1}{dt}\Big(\sum_{i=1}^{N}q_i * k_1\cos\omega t\Big)\\ &= 0 + \frac{1}{dt}\Big(\sum_{i=1}^{N}q_i k_1\cos\omega t\Big)\\ &= \Big(\sum_{i=1}^{N}-\omega q_i k_1\sin\omega t\Big) \neq 0\end{aligned} \tag{34}$$

So it is non-conservative.

**Verify the systems' time-reversal symmetry** We do the transformation $R \; : \; (\mathbf{q},\mathbf{p},t) \mapsto (\mathbf{q},-\mathbf{p},-t)$.

$$\mathcal{H}(\mathbf{q},\mathbf{p}) = \sum_{i=1}^{N}\frac{\mathbf{p_i}^2}{2m_i} + \frac{1}{2}\sum_{i=1}^{N}\sum_{j\in N_i}^{N}\frac{1}{2}k(\mathbf{q_i}-\mathbf{q_j})^2 + \sum_{i=1}^{N}q_i * k_1\cos\omega t,$$

$$\mathcal{H}(\mathbf{q},-\mathbf{p}) = \sum_{i=1}^{N}\frac{(-\mathbf{p_i})^2}{2m_i} + \frac{1}{2}\sum_{i=1}^{N}\sum_{j\in N_i}^{N}\frac{1}{2}k(\mathbf{q_i}-\mathbf{q_j})^2 + \sum_{i=1}^{N}q_i * k_1\cos\omega(-t), \tag{35}$$

It is obvious $\mathcal{H}(\mathbf{q},\mathbf{p},t) = \mathcal{H}(\mathbf{q},-\mathbf{p},t)$, so it is reversible

### B.3 NON-CONSERVATIVE AND IRREVERSIBLE SYSTEMS.

A simple example is an n-body spring system with frictions proportional to its velocity, $\gamma$ is the coefficient of friction, which can be described by:

$$
\begin{aligned}
\frac{d\mathbf{q}_i}{dt} &= \frac{\mathbf{p}_i}{m} \\
\frac{d\mathbf{p}_i}{dt} &= -k_0\mathbf{q}_i - \gamma\frac{\mathbf{p}_i}{m}
\end{aligned}
\tag{36}
$$

The Hamilton's equations are:

$$
\begin{aligned}
\frac{\partial \mathcal{H}(\mathbf{q},\mathbf{p})}{\partial \mathbf{p_i}} &= \frac{d\mathbf{q_i}}{dt} = \frac{\mathbf{p_i}}{m} \\
\frac{\partial \mathcal{H}(\mathbf{q},\mathbf{p})}{\partial \mathbf{q_i}} &= -\frac{d\mathbf{p_i}}{dt} = \sum_{j\in N_i} k(\mathbf{q_i}-\mathbf{q_j}) + \gamma\frac{\mathbf{p_i}}{m}
\end{aligned}
\tag{37}
$$

Hence, we can obtain the Hamiltonian through the integration of the above equation:

$$
\mathcal{H}(\mathbf{q},\mathbf{p}) = \sum_{i=1}^{N} \frac{\mathbf{p_i}^2}{2m_i} + \frac{1}{2}\sum_{i=1}^{N}\sum_{j\in N_i}^{N} \frac{1}{2}k(\mathbf{q_i}-\mathbf{q_j})^2 + \sum_{i=1}^{N}\frac{\gamma}{m}\int_0^t \frac{\mathbf{p_i}^2}{m}dt,
\tag{38}
$$

**Verify the systems' energy conservation**

$$
\begin{aligned}
&\frac{d\mathcal{H}(\mathbf{q},\mathbf{p})}{dt} \\
=&\frac{1}{dt}\left(\sum_{i=1}^{N}\frac{\mathbf{p_i}^2}{2m_i}\right) + \frac{1}{dt}\left(\frac{1}{2}\sum_{i=1}^{N}\sum_{j\in N_i}^{N}\frac{1}{2}k(\mathbf{q_i}-\mathbf{q_j})^2\right) + \frac{1}{dt}\left(\sum_{i=1}^{N}\frac{\gamma}{m}\int_0^t\frac{\mathbf{p_i}^2}{m}dt\right) \\
=&0 + \frac{1}{dt}\left(\sum_{i=1}^{N}\frac{\gamma}{m}\int_0^t\frac{\mathbf{p_i}^2}{m}dt\right) \\
=&\left(\sum_{i=1}^{N}\frac{\gamma}{m}\frac{\mathbf{p_i}^2}{m}\right) \neq 0
\end{aligned}
\tag{39}
$$

So it is non-conservative.

**Verify the systems' time-reversal symmetry** We do the transformation $R : (\mathbf{q},\mathbf{p},t) \mapsto (\mathbf{q},-\mathbf{p},-t)$.

$$
\begin{aligned}
\mathcal{H}(\mathbf{q},\mathbf{p}) &= \sum_{i=1}^{N}\frac{\mathbf{p_i}^2}{2m_i} + \frac{1}{2}\sum_{i=1}^{N}\sum_{j\in N_i}^{N}\frac{1}{2}k(\mathbf{q_i}-\mathbf{q_j})^2 + \sum_{i=1}^{N}\frac{\gamma}{m}\int_0^t\frac{\mathbf{p_i}^2}{m}dt, \\
\mathcal{H}(\mathbf{q},-\mathbf{p}) &= \sum_{i=1}^{N}\frac{(-\mathbf{p_i})^2}{2m_i} + \frac{1}{2}\sum_{i=1}^{N}\sum_{j\in N_i}^{N}\frac{1}{2}k(\mathbf{q_i}-\mathbf{q_j})^2 + \sum_{i=1}^{N}\frac{\gamma}{m}\int_0^{(-t)}\frac{\mathbf{p_i}^2}{m}d(-t),
\end{aligned}
\tag{40}
$$

It is obvious $\mathcal{H}(\mathbf{q},\mathbf{p},t) \neq \mathcal{H}(\mathbf{q},-\mathbf{p},t)$, so it is irreversible

## C DATASET

In our experiments, all datasets are synthesized from ground-truth physical law via sumulation. We generate four simulated datasets: three $n$-body spring systems under damping, periodic, or no external force, and one chaotic tripe pendulum dataset with three sequentially connected stiff sticks that form. We name the first three as *Sipmle Spring*, *Forced Spring*, and *Damped Spring* respectively. All $n$-body spring systems contain 5 interacting balls, with varying connectivities. Each *Pendulum* system contains 3 connected stiff sticks.

For the $n$-body spring system, we randomly sample whether a pair of objects are connected, and model their interaction via forces defined by Hookes law. In the *Damped spring*, the objects have

an additional friction force that is opposite to their moving direction and whose magnitude is proportional to their speed. In the *Forced spring*, all objects have the same external force that changes direction periodically. We show in Figure 1(a), the energy variation in both of the *Damped spring* and *Forced spring* is significant.

For the chaotic triple *Pendulum* , the equations governing the motion are inherently nonlinear. Although this system is deterministic, it is also highly sensitive to the initial condition and numerical errors (Shinbrot et al., 1992; Awrejcewicz et al., 2008; Stachowiak & Okada, 2006). This property is often referred to as the "butterfly effect", as depicted in Fig 7. Unlike for $n$-body spring systems, where the forces and equations of motion can be easily articulated, for the *Pendulum*, the explicit forces cannot be directly defined, and the motion of objects can only be described through Lagrangian formulations North (2021), making the modeling highly complex and raising challenges for accurate learning.

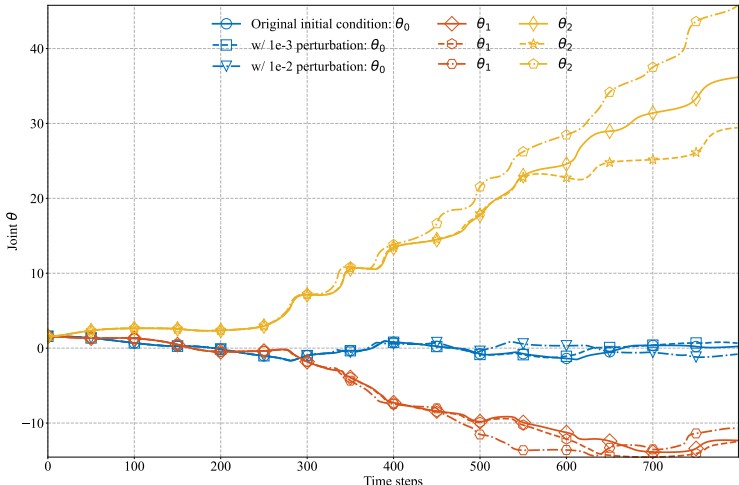

Figure 7: Illustration to show the pendulum is highly-sensitive to initial states

We simulate the trajectories by using Euler's method for $n$-body spring systems and using the 4th order Runge-Kutta (RK4) method for the *Pendulum*. We integrate with a fixed step size and subsample every 100 steps. For training, we use a total of 6000 forward steps. To generate irregularly sampled partial observations, we follow Huang et al. (2020) and sample the number of observations n from a uniform distribution U(40, 52) and draw the n observations uniformly for each object. For testing, we additionally sample 40 observations following the same procedure from PDE steps [6000, 12000], besides generating observations from steps [1, 6000]. The above sampling procedure is conducted independently for each object. We generate 20k training samples and 5k testing samples for each dataset. The features (position/velocity) are normalized to the maximum absolute value of 1 across training and testing datasets.

In the following subsections, we show the dynamical equations of each dataset in detail.

## C.1    SPRING

### C.1.1    SIMPLE SPRING

The dynamical equations of *simple spring* are as follows:

$$\frac{d\mathbf{q_i}}{dt} = \frac{\mathbf{p_i}}{m}$$
$$\frac{d\mathbf{p_i}}{dt} = \sum_{j \in N_i}^{N} -k(\mathbf{q_i} - \mathbf{q_j}) \tag{41}$$

where where $\mathbf{q} = (\mathbf{q_1}, \mathbf{q_2}, \cdots, \mathbf{q_N})$ is a set of positions of each object , $\mathbf{p} = (\mathbf{p_1}, \mathbf{p_2}, \cdots, \mathbf{p_N})$ is a set of momenta of each object. We set the mass of each object $m = 1$, the spring constant $k = 0.1$.

### C.1.2 DAMPED SPRING

The dynamical equations of *damped spring* are as follows:

$$\frac{d\mathbf{q_i}}{dt} = \frac{\mathbf{p_i}}{m}$$
$$\frac{d\mathbf{p_i}}{dt} = \sum_{j \in N_i} -k(\mathbf{q_i} - \mathbf{q_j}) - \gamma \frac{\mathbf{p}_i}{m} \tag{42}$$

where where $\mathbf{q} = (\mathbf{q_1}, \mathbf{q_2}, \cdots, \mathbf{q_N})$ is a set of positions of each object, $\mathbf{p} = (\mathbf{p_1}, \mathbf{p_2}, \cdots, \mathbf{p_N})$ is a set of momenta of each object, We set the mass of each object $m = 1$, the spring constant $k = 0.1$, the coefficient of friction $\gamma = 10$.

### C.1.3 FORCED SPRING

The dynamical equations of *forced spring* system are as follows:

$$\frac{d\mathbf{q_i}}{dt} = \frac{\mathbf{p_i}}{m}$$
$$\frac{d\mathbf{p_i}}{dt} = \sum_{j \in N_i}^{N} -k(\mathbf{q_i} - \mathbf{q_j}) - k_1 \cos \omega t, \tag{43}$$

where where $\mathbf{q} = (\mathbf{q_1}, \mathbf{q_2}, \cdots, \mathbf{q_N})$ is a set of positions of each object , $\mathbf{p} = (\mathbf{p_1}, \mathbf{p_2}, \cdots, \mathbf{p_N})$ is a set of momenta of each object. We set the mass of each object $m = 1$ , the spring constant $k = 0.1$, the external strength $k_1 = 10$ and the frequency of variation $\omega = 1$

We simulate the positions and momentums of three spring systems by using Euler methods as follows:

$$\mathbf{q_i}(t+1) = \mathbf{q_i}(t) + \frac{d\mathbf{q_i}}{dt} \Delta t$$
$$\mathbf{p_i}(t+1) = \mathbf{p_i}(t) + \frac{d\mathbf{p_i}}{dt} \Delta t \tag{44}$$

where $\frac{d\mathbf{q_i}}{dt}$ and $\frac{d\mathbf{p_i}}{dt}$ were defined as above for each datasets, and $\Delta t = 0.001$ is the integration steps.

### C.2 CHAOTIC PENDULUM

In this section, we demonstrate how to derive the dynamics equations for a chaotic triple pendulum using the Lagrangian formalism.

The moment of inertia of each stick about the centroid is

$$I = \frac{1}{12} m l^2 \tag{45}$$

The position of the center of gravity of each stick is as follows:

$$x_1 = \frac{l}{2} \sin \theta_1, \quad y_1 = -\frac{l}{2} \cos \theta_1$$
$$x_2 = l(\sin \theta_1 + \frac{1}{2} \sin \theta_2), \quad y_2 = -l(\cos \theta_1 + \frac{1}{2} \cos \theta_2) \tag{46}$$
$$x_3 = l(\sin \theta_1 + \sin \theta_2 + \frac{1}{2} \sin \theta_3), \quad y_3 = -l(\cos \theta_1 + \cos \theta_2 + \frac{1}{2} \cos \theta_3)$$

The change in the center of gravity of each stick is:

$$\dot{x}_1 = \frac{l}{2} \cos \theta_1 \cdot \dot{\theta}_1, \quad \dot{y}_1 = \frac{l}{2} \sin \theta_1 \cdot \dot{\theta}_1$$
$$\dot{x}_2 = l(\cos \theta_1 \cdot \dot{\theta}_1 + \frac{1}{2} \cos \theta_2 \cdot \dot{\theta}_2), \quad \dot{y}_2 = l(\sin \theta_1 \cdot \dot{\theta}_1 + \frac{1}{2} \sin \theta_2 \cdot \dot{\theta}_2) \tag{47}$$
$$\dot{x}_3 = l(\cos \theta_1 \cdot \dot{\theta}_1 + \cos \theta_2 \cdot \dot{\theta}_2 + \frac{1}{2} \cos \theta_3 \cdot \dot{\theta}_3), \quad \dot{y}_3 = l(\sin \theta_1 \cdot \dot{\theta}_1 + \sin \theta_2 \cdot \dot{\theta}_2 + \frac{1}{2} \sin \theta_3 \cdot \dot{\theta}_3)$$

The Lagrangian L of this triple pendulum system is:

$$
\begin{aligned}
\mathcal{L} =& T - V \\
=& \frac{1}{2}m(\dot{x_1}^2 + \dot{x_2}^2 + \dot{x_3}^2 + \dot{y_1}^2 + \dot{y_2}^2 + \dot{y_3}^2) + \frac{1}{2}I(\dot{\theta_1}^2 + \dot{\theta_2}^2 + \dot{\theta_3}^2) - mg(y_1 + y_2 + y_3) \\
=& \frac{1}{6}ml(9\dot{\theta_2}\dot{\theta_1}l\cos(\theta_1 - \theta_2) + 3\dot{\theta_3}\dot{\theta_1}l\cos(\theta_1 - \theta_3) + 3\dot{\theta_2}\dot{\theta_3}l\cos(\theta_2 - \theta_3) + 7\dot{\theta_1^2}l + 4\dot{\theta_2^2}l + \dot{\theta_3^2}l \\
& + 15g\cos(\theta_1) + 9g\cos(\theta_2) + 3g\cos(\theta_3))
\end{aligned}
\tag{48}
$$

The Lagrangian equation is defined as follows:

$$
\frac{d}{dt}\frac{\partial\mathcal{L}}{\partial\dot{\theta}} - \frac{\partial\mathcal{L}}{\partial\theta} = 0
\tag{49}
$$

and we also have:

$$
\begin{aligned}
\frac{\partial\mathcal{L}}{\partial\dot{\theta}} &= \frac{\partial T}{\partial\dot{\theta}} = p \\
\dot{p} &= \frac{d}{dt}\frac{\partial\mathcal{L}}{\partial\dot{\theta}} = \frac{\partial\mathcal{L}}{\partial\theta}
\end{aligned}
\tag{50}
$$

where p is the Angular Momentum.
We can list the equations for each of the three sticks separately:

$$
\begin{aligned}
p_1 &= \frac{\partial\mathcal{L}}{\partial\dot{\theta_1}} \quad \dot{p_1} = \frac{\partial\mathcal{L}}{\partial\theta_1} \\
p_2 &= \frac{\partial\mathcal{L}}{\partial\dot{\theta_2}} \quad \dot{p_2} = \frac{\partial\mathcal{L}}{\partial\theta_2} \\
p_3 &= \frac{\partial\mathcal{L}}{\partial\dot{\theta_3}} \quad \dot{p_3} = \frac{\partial\mathcal{L}}{\partial\theta_3}
\end{aligned}
\tag{51}
$$

Finally, we have :

$$
\begin{cases}
\dot{\theta_1} = \frac{6(9p_1\cos(2(\theta_2-\theta_3))+27p_2\cos(\theta_1-\theta_2)-9p_2\cos(\theta_1+\theta_2-2\theta_3)+21p_3\cos(\theta_1-\theta_3)-27p_3\cos(\theta_1-2\theta_2+\theta_3)-23p_1)}{ml^2(81\cos(2(\theta_1-\theta_2))-9\cos(2(\theta_1-\theta_3))+45\cos(2(\theta_2-\theta_3))-169)} \\
\dot{\theta_2} = \frac{6(27p_1\cos(\theta_1-\theta_2)-9p_1\cos(\theta_1+\theta_2-2\theta_3)+9p_2\cos(2(\theta_1-\theta_3))-27p_3\cos(2\theta_1-\theta_2-\theta_3)+57p_3\cos(\theta_2-\theta_3)-47p_2)}{ml^2(81\cos(2(\theta_1-\theta_2))-9\cos(2(\theta_1-\theta_3))+45\cos(2(\theta_2-\theta_3))-169)} \\
\dot{\theta_3} = \frac{6(21p_1\cos(\theta_1-\theta_3)-27p_2\cos(2\theta_1-\theta_2+\theta_3)-27p_2\cos(2\theta_1-\theta_2-\theta_3)+57p_2\cos(\theta_2-\theta_3)+81p_3\cos(2(\theta_1-\theta_2))-143p_3)}{ml^2(81\cos(2(\theta_1-\theta_2))-9\cos(2(\theta_1-\theta_3))+45\cos(2(\theta_2-\theta_3))-169)} \\
\dot{p_1} = -\frac{1}{2}ml\left(3\dot{\theta_2}\dot{\theta_1}l\sin(\theta_1-\theta_2) + \dot{\theta_1}\dot{\theta_3}l\sin(\theta_1-\theta_3) + 5g\sin(\theta_1)\right) \\
\dot{p_1} = -\frac{1}{2}ml\left(-3\dot{\theta_1}\dot{\theta_2}l\sin(\theta_1-\theta_2) + \dot{\theta_2}\dot{\theta_3}l\sin(\theta_2-\theta_3) + 3g\sin(\theta_2)\right) \\
\dot{p_1} = -\frac{1}{2}ml\left(\dot{\theta_1}\dot{\theta_3}l\sin(\theta_1-\theta_3) + \dot{\theta_2}\dot{\theta_3}l\sin(\theta_2-\theta_3) - g\sin(\theta_3)\right)
\end{cases}
\tag{52}
$$

We simulate the angular of the three sticks by using the Runge-Kutta 4th Order Method as follows:

$$
\begin{aligned}
\Delta\boldsymbol{\theta}^1(t) &= \dot{\boldsymbol{\theta}}(t, \boldsymbol{\theta}(t)) \cdot \Delta t \\
\Delta\boldsymbol{\theta}^2(t) &= \dot{\boldsymbol{\theta}}(t + \frac{\Delta t}{2}, \boldsymbol{\theta}(t) + \frac{\Delta\boldsymbol{\theta}^1(t)}{2}) \cdot \Delta t \\
\Delta\boldsymbol{\theta}^3(t) &= \dot{\boldsymbol{\theta}}(t + \frac{\Delta t}{2}, \boldsymbol{\theta}(t) + \frac{\Delta\boldsymbol{\theta}^2(t)}{2}) \cdot \Delta t \\
\Delta\boldsymbol{\theta}^4(t) &= \dot{\boldsymbol{\theta}}(t + \Delta t, \boldsymbol{\theta}(t) + \Delta\boldsymbol{\theta}^3(t)) \cdot \Delta t \\
\Delta\boldsymbol{\theta}(t) &= \frac{1}{6}(\Delta\boldsymbol{\theta}^1(t) + \Delta\boldsymbol{\theta}^2(t) + \Delta\boldsymbol{\theta}^3(t) + \Delta\boldsymbol{\theta}^4(t)) \\
\boldsymbol{\theta}(t+1) &= \boldsymbol{\theta}(t) + \Delta\boldsymbol{\theta}(t)
\end{aligned}
\tag{53}
$$

where $\dot{\boldsymbol{\theta}}$ was defined as above , and $\Delta t = 0.0001$ is the integration steps.

## D  MODEL DETAILS

In the following we introduce in details how we implement our model and each baseline.

### D.1 INITIAL STATE ENCODER

The initial state encoder computes the latent node initial states $z_i(t)$ for all agents simultaneously considering their mutual interaction. Specifically, it first fuses all observations into a temporal graph and conducts dynamic node representation through a spatial-temporal GNN as in Huang et al. (2020):

$$h_{j(t)}^{l+1} = h_{j(t)}^l + \sigma\left(\sum_{i(t') \in \mathcal{N}_{j(t)}} \alpha_{i(t') \to j(t)}^l \times W_v \hat{h}_{i(t')}^{l-1}\right)$$

$$\alpha_{i(t') \to j(t)}^l = \left(W_k \hat{h}_{i(t')}^{l-1}\right)^T \left(W_q h_{j(t)}^{l-1}\right) \cdot \frac{1}{\sqrt{d}}, \quad \hat{h}_{i(t')}^{l-1} = h_{i(t')}^{l-1} + \text{TE}(t' - t) \qquad (54)$$

$$\text{TE}(\Delta t)_{2i} = \sin\left(\frac{\Delta t}{10000^{2i/d}}\right), \quad \text{TE}(\Delta t)_{2i+1} = \cos\left(\frac{\Delta t}{10000^{2i/d}}\right),$$

where $||$ denotes concatenation; $\sigma(\cdot)$ is a non-linear activation function; $d$ is the dimension of node embeddings. The node representation is computed as a weighted summation over its neighbors plus residual connection where the attention score is a transformer-based Vaswani et al. (2017) dot-product of node representations by the use of value, key, query projection matrices $W_v, W_k, W_q$. Here $h_{j(t)}^l$ is the representation of agent $j$ at time $t$ in the $l$-th layer. $i(t')$ is the general index for neighbors connected by temporal edges (where $t' \neq t$) and spatial edges (where $t = t'$ and $i \neq j$). The temporal encoding Hu et al. (2020) is added to a neighborhood node representation in order to distinguish its message delivered via spatial and temporal edges. Then, we stack $L$ layers to get the final representation for each observation node: $h_i^t = h_{i(t)}^L$. Finally, we employ a self-attention mechanism to generate the sequence representation $u_i$ for each agent as their latent initial states:

$$u_i = \frac{1}{K}\sum_t \sigma\left(a_i^T \hat{h}_i^t \hat{h}_i^t\right), \quad a_i = \tanh\left(\left(\frac{1}{K}\sum_t \hat{h}_i^t\right) W_a\right), \qquad (55)$$

where $a_i$ is the average of observation representations with a nonlinear transformation $W_a$ and $\hat{h}_i^t = h_i^t + \text{TE}(t)$. $K$ is the number of observations for each trajectory. Compared with recurrent models such as RNN, LSTM Sepp & Jürgen (1997), it offers better parallelization for accelerating training speed and in the meanwhile alleviates the vanishing/exploding gradient problem brought by long sequences.

Given the latent initial states, the dynamics of the whole system are determined by the ODE function $g$ which we parametrize as a GNN as in Huang et al. (2020) to capture the continuous interaction among agents. We then employ Multilayer Perceptron (MLP) as a decoder to predict the trajectories $\hat{y}_i(t)$ from the latent states $z_i(t)$.

$$z_i(t), z_i(1), z_i(2)\cdots z_i(T) = \text{ODEsolver}(g, [z_1(0), z_2(0)\cdots z_N(0)], (t_0, t_1\cdots t_T))$$

$$\hat{y}_i(t) = f_{dec}(z_i(t)) \qquad (56)$$

### D.2 IMPLEMENTATION DETAILS

**TANGO**

Our implementation of TANGO follows GraphODE pipeline. We implement the initial state encoder using a 2-layer GNN with a hidden dimension of 64 across all datasets. We use ReLU for nonlinear activation. For the sequence self-attention module, we set the output dimension to 128. The encoder's output dimension is set to 16, and we add 64 additional dimensions initialized with all zeros to the latent states $z_i(t)$ to stabilize the training processes as in Huang et al. (2021). The GNN ODE function is implemented with a single-layer GNN from Kipf et al. (2018) with hidden dimension 128. To compute trajectories, we use the Runge-Kutta method from torchdiffeq python package s(Chen et al., 2021) as the ODE solver and a one-layer MLP as the decoder.

We implement our model in pytorch. Encoder, generative model, and the decoder parameters are jointly optimized with AdamW optimizer (Loshchilov & Hutter, 2019) using a learning rate of 0.0001 for spring datasets and 0.00001 for *Pendulum*. The batch size for all datasets is set to 512.

TANGO$_{\mathcal{L}_{rev}=\text{gt-rev}}$ and TANGO$_{\mathcal{L}_{rev}=\text{rev2}}$ share the same architecture and hyparameters as TANGO, with different implementations of the loss function. In TANGO$_{\mathcal{L}_{rev}=\text{gt-rev}}$, instead of comparing forward and reverse trajectories, we look at the L2 distance between the ground truth and reverse trajectories when computing the reversal loss.

For TANGO$_{\mathcal{L}_{rev}=\text{rev2}}$, we implement the reversal loss following Huh et al. (2020) with one difference: we do not apply the reverse operation to the momentum portion of the initial state to the ODE function. This is because the initial hidden state is an output of the encoder that mixes position and momentum information. Note that we also remove the additional dimensions to the latent state that TANGO has.

**LatentODE**

We implement the Latent ODE sequence to sequence model as specified in Rubanova et al. (2019). We use a 4-layer ODE function in the recognition ODE, and a 2-layer ODE function in the generative ODE. The recognition and generative ODEs use Euler and Dopri5 as solvers (Chen et al., 2021), respectively. The number of units per layer is 1000 in the ODE functions and 50 in GRU update networks. The dimension of the recognition model is set to 100. The model is trained with a learning rate of 0.001 with an exponential decay rate of 0.999 across different experiments. Note that since latentODE is a single-agent model, we compute the trajectory of each object independently when applying it to multi-agent systems.

**HODEN**

To adapt HODEN, which requires full initial states of all objects, to systems with partial observations, we compute each objects initial state via linear spline interpolation if it is missing. Following the setup in Huh et al. (2020), we have two 2-layer linear networks with Tanh activation in between as ODE functions, in order to model both positions and momenta. Each network has a 1000-unit layer followed by a single-unit layer. The model is trained with a learning rate of 0.00001 using a cosine scheduler.

**TRS-ODEN**

Similar to HODEN, we compute each objects initial state via linear spline interpolation if it is missing. As in Huh et al. (2020), we use a 2-layer linear network with Tanh activation in between as the ODE functions, and the Leapfrog method for solving ODEs. The network has 1000 hidden units and is trained with a learning rate of 0.00001 using a cosine scheduler.

**TRS-ODEN$_{\text{GNN}}$**

For TRSODEN$_{\text{GNN}}$, we substitute the ODE function in TRS-ODEN with a GraphODE network. The GraphODE generative model is implemented with a single-layer GNN with hidden dimension 128. As in HODEN and TRS-ODEN, we compute each objects missing initial state via linear spline interpolation and the Leapfrog method for solving ODE. For all datasets, we use 0.5 as the coefficient for the reversal loss in Huh et al. (2020), and 0.0002 as the learning rate under cosine scheduling.

**LGODE**

Our implementation follows Huang et al. (2020) except we remove the Variational Autoencoder (VAE) from the initial state encoder. Instead of using the output from the encoder GNN as the mean and std of the VAE, we directly use it as the latent initial state. That is, the initial states are deterministic instead of being sampled from a distribution. We use the same architecture as in TANGO and train the model using an AdamW optimizer with a learning rate of 0.0001 across all datasets.

# E   ADDITIONAL EXPERIMENTS

## E.1   RESULTS ON MUJOCO DATASET.

In addition to assessing our model's performance on simulated datasets, we also conduct experiments using a real-world motion capture datasetCMU (2003). In particular, we focus on the walking sequences of subject 35. Each sample in this dataset is represented by 31 trajectories, each corresponding to the movement of a single joint. For each joint, we first randomly sample the number

Table 3: Evaluation results on MSE ( $10^{-2}$) on MoJoCo Dataset

|  | LatentODE | HODEN | TRS-ODEN$_{\text{GNN}}$ | LGODE | TANGO |
|---|---|---|---|---|---|
| MSE | 2.9061 | 1.9855 | 0.2609 | 0.7610 | **0.2192** |

Table 4: Evaluation results on MSE ($10^{-2}$) over different solvers.

| Dataset | *Simple Spring* | | *Forced Spring* | | *Damped Spring* | | *Pendulum* | |
|---|---|---|---|---|---|---|---|---|
| Solvers | Euler | RK4 | Euler | RK4 | Euler | RK4 | Euler | RK4 |
| LGODE | 1.8443 | 1.7429 | 2.0462 | 1.8929 | 1.1686 | 0.9718 | 1.4634 | 1.4156 |
| TANGO | 1.4864 | 1.1178 | 1.6058 | 1.4525 | 0.8070 | 0.5944 | 1.3093 | 1.2527 |
| % Improvement | 19.4057 | 35.8655 | 21.5228 | 23.2659 | 30.9430 | 38.8352 | 10.5303 | 11.5075 |

of observations from a uniform distribution $\mathcal{U}(30, 42)$ and then sample uniformly from the first 50 frames for training and validation trajectories. For testing, we additionally sampled 40 observations from frames $[51, 99]$. As shown in Table 3, TANGO consistently outperforms existing baselines.

### E.2 COMPARISON OF DIFFERENT SOLVERS

We next show our model's sensitivity regarding solvers with different precisions. Specifically, we compare against Euler and Runge-Kutta (RK4) where the latter is a higher-precision solver. We show the comparison against LGODE and TANGO in Table 4.

We can firstly observe that TANGO consistently outperforms LGODE, which is our strongest baseline across different solvers and datasets, indicating the effectiveness of the proposed time-reversal symmetry loss. Secondly, we compute the improvement ratio as $\frac{LGODE - TANGO}{LGODE}$. We can see that the improvement ratios get larger when using RK4 over Euler. This can be understood as our reversal loss is minimizing higher-order Tayler expansion terms (Theorem 1) thus compensating numerical errors brought by ODE solvers.

### E.3 REVERSAL LOSS VISUALIZATIONS

To further illustrate the issue of energy explosion from LG-ODE that is purely data-driven, we visualize the reversal loss over training epochs from LG-ODE[6] and TANGO in Figure 8. As results suggest, LG-ODE is having increased reversal loss over training epochs, meaning it is violating the time-reversal symmetry sharply, as contrast to TANGO which has decreased reversal loss over epochs.

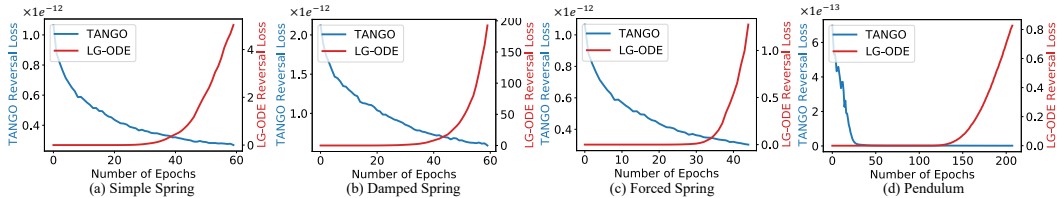

Figure 8: Time-Reversal symmetry loss visualization across datasets.

## F DISCUSSION ABOUT REVERSIBLE NEURAL NETWORKS

In literature, there is another line of research about building reversible neural networks (NNs). For example, Chang et al. (2018) formulates three architectures for reversible neural networks to address the stability issue and achieve arbitrary deep lengths, motivated by dynamical system modeling. Liu et al. (2019) employs normalizing flow to create a generative model of graph structures. They all propose novel architectures to construct reversible NN where intermediate states across layer depths do not need to be stored, thus improving memory efficiency.

---

[6]There is no reversal loss backpropagation in LG-ODE, we just compute its value along training.

However, we'd like to clarify that reversible NNs (RevNet) do not resolve the time-reversal symmetry problem that we're studying. The core of RevNet is that input can be recovered from output via a reversible operation (which is another operator), similar as any linear operator $W(\cdot)$ have a reversed projector $W^{-1}(\cdot)$. In the contrary, what we want to study is that the **same operator** can be used for both forward and backward prediction over time, and keep the trajectory the same. That being said, to generate the forward and backward trajectories, we are using the same $g(\cdot)$, instead of $g(\cdot), g^{-1}(\cdot)$ respectively.

In summary, though both reversible NN and time-reversal symmetry share similar insights and intuition, they're talking about different things: reversible NNs make every operator $g(\cdot)$ having a $g^{-1}(\cdot)$, while time-reversible assume the trajectory get from $\hat{z}^{fwd} = g(z)$ and $\hat{z}^{bwd} = -g(z)$ to be closer. Making $g$ to be reversible cannot make the system to be time-reversible.

