$z^{\mathrm{gt}}(t_j)$ by $z_j^{\mathrm{gt}}$, $z^{\mathrm{fwd}}(t_j)$ by $z_j^{\mathrm{fwd}}$ and $z^{\mathrm{rev}}(t_j)$ by $z_j^{\mathrm{rev}}$, and we will use Mathematical Induction to prove the theorem.

### A.2.1  RECONSTRUCTION LOSS ($\mathcal{L}_{pred}$) ANALYSIS.

First, we bound the forward loss $\sum_{j=0}^T \|z_j^{\mathrm{fwd}} - z_j^{\mathrm{gt}}\|_2^2$. Since our method models the momentum and position of the system, we can write the following Taylor expansion of the forward process, where for any $0 \le j < T$:

$$
\begin{cases}
q_{j+1}^{\mathrm{fwd}} = q_j^{\mathrm{fwd}} + (p_j^{\mathrm{fwd}}/m)\Delta t + (\dot{p}_j^{\mathrm{fwd}}/2m)\Delta t^2 + \mathcal{O}(\Delta t^3), & \text{(16a)} \\
p_{j+1}^{\mathrm{fwd}} = p_j^{\mathrm{fwd}} + \dot{p}_j^{\mathrm{fwd}}\Delta t + \mathcal{O}(\Delta t^2), & \text{(16b)} \\
\dot{p}_{j+1}^{\mathrm{fwd}} = \dot{p}_j^{\mathrm{fwd}} + \mathcal{O}(\Delta t), & \text{(16c)}
\end{cases}
$$

and for the ground truth process, we also have from Taylor expansion that

$$
\begin{cases}
q_{j+1}^{\mathrm{gt}} = q_j^{\mathrm{gt}} + (p_j^{\mathrm{gt}}/m)\Delta t + (\dot{p}_j^{\mathrm{gt}}/2m)\Delta t^2 + \mathcal{O}(\Delta t^3), & \text{(17a)} \\
p_{j+1}^{\mathrm{gt}} = p_j^{\mathrm{gt}} + \dot{p}_j^{\mathrm{gt}}\Delta t + \mathcal{O}(\Delta t^2), & \text{(17b)} \\
\dot{p}_{j+1}^{\mathrm{gt}} = \dot{p}_j^{\mathrm{gt}} + \mathcal{O}(\Delta t). & \text{(17c)}
\end{cases}
$$

With these, we aim to prove that for any $k = 0, 1, \cdots, T$, the following hold :

$$
\begin{cases}
\|q_k^{\mathrm{fwd}} - q_k^{\mathrm{gt}}\|_2 \le C_2^{\mathrm{fwd}} k^2 \Delta t^2, & \text{(18a)} \\
\|p_k^{\mathrm{fwd}} - p_k^{\mathrm{gt}}\|_2 \le C_1^{\mathrm{fwd}} k \Delta t, & \text{(18b)}
\end{cases}
$$

where $C_1^{\mathrm{fwd}}$ and $C_2^{\mathrm{fwd}}$ are constants.

**Base Case** $k = 0$**:** Based on the initialization rules, it is obvious that $\left\|q_0^{\text{fwd}} - q_0^{\text{gt}}\right\|_2 = 0$ and $\left\|p_0^{\text{fwd}} - p_0^{\text{gt}}\right\|_2 = 0$, thus (18a) and (18b) both hold for $k = 0$.

**Inductive Hypothesis:** Assume (18a) and (18b) hold for $k = j$, which means:

$$\begin{cases} \|q_j^{\text{fwd}} - q_j^{\text{gt}}\|_2 \le C_2^{\text{fwd}} j^2 \Delta t^2, & \text{(19a)} \\ \|p_j^{\text{fwd}} - p_j^{\text{gt}}\|_2 \le C_1^{\text{fwd}} j \Delta t, & \text{(19b)} \end{cases}$$

**Inductive Proof:** We need to prove (18a) and (18b) hold for $k = j + 1$.

First, using (16c) and (17c), we have

$$\left\|\dot{p}_{j+1}^{\text{fwd}} - \dot{p}_{j+1}^{\text{gt}}\right\|_2 = \left\|\dot{p}_j^{\text{fwd}} - \dot{p}_j^{\text{gt}}\right\|_2 + \mathcal{O}(\Delta t) = \left\|\dot{p}_0^{\text{fwd}} - \dot{p}_0^{\text{gt}}\right\|_2 + \mathcal{O}\big((j+1)\Delta t\big) = \mathcal{O}(1), \quad \text{(20)}$$

where we iterate through $j, j-1, \cdots, 0$ in the second equality. Then using (17b) and (16b), we get for $j+1$ that

$$\begin{aligned} \left\|p_{j+1}^{\text{fwd}} - p_{j+1}^{\text{gt}}\right\|_2 &= \left\|\big(p_j^{\text{fwd}} + \dot{p}_j^{\text{fwd}}\Delta t\big) - \big(p_j^{\text{gt}} + \dot{p}_j^{\text{gt}}\Delta t\big) + \mathcal{O}(\Delta t^2)\right\|_2 \\ &\le \left\|p_j^{\text{fwd}} - p_j^{\text{gt}}\right\|_2 + \left\|\dot{p}_j^{\text{fwd}} - \dot{p}_j^{\text{gt}}\right\|_2 \Delta t + \mathcal{O}(\Delta t^2) \\ &\le \big[C_1^{\text{fwd}} j + \mathcal{O}(1)\big]\Delta t, \end{aligned}$$

where the first inequality uses the triangle inequality, and in the second inequality we use (19b) as well as (20). We can see there exists $C_1^{\text{fwd}}$ such that the final expression above is upper bounded by $C_1^{\text{fwd}}(j+1)\Delta t$, with which the claim holds for $j+1$.

Next for (18a), using (17a) and (16a), we get for any $j$ that

$$\begin{aligned} \left\|q_{j+1}^{\text{fwd}} - q_{j+1}^{\text{gt}}\right\|_2 &= \left\|\big(q_j^{\text{fwd}} + (p_j^{\text{fwd}}/m)\Delta t + (\dot{p}_j^{\text{fwd}}/2m)\Delta t^2\big)\right. \\ &\qquad \left. - \big(q_j^{\text{gt}} + (p_j^{\text{gt}}/m)\Delta t + (\dot{p}_j^{\text{gt}}/2m)\Delta t^2\big) + \mathcal{O}(\Delta t^3)\right\|_2 \\ &\le \left\|q_j^{\text{fwd}} - q_j^{\text{gt}}\right\|_2 + \frac{1}{m}\left\|p_j^{\text{fwd}} - p_j^{\text{gt}}\right\|_2 \Delta t + \frac{1}{2m}\left\|\dot{p}_j^{\text{fwd}} - \dot{p}_j^{\text{gt}}\right\|_2 \Delta t^2 + \mathcal{O}(\Delta t^3) \\ &\le \left[C_2^{\text{fwd}} j^2 + \frac{C_1^{\text{fwd}}}{m} j + \mathcal{O}(1)\right]\Delta t^2, \end{aligned}$$

where the first inequality uses the triangle inequality, and in the second inequality we use (19a) and (19b) as well as (20). Thus with an appropriate $C_2^{\text{fwd}}$, we have the final expression above is upper bounded by $C_2^{\text{fwd}}(j+1)^2 \Delta t^2$, and so the claim holds for $j+1$.

Since both the base case and the inductive step have been proven, by the principle of mathematical induction, (18a) and (18b) holds for all $k = 0, 1, \cdots, T$.

With this, we finish the forward proof by plugging (18a) and (18b) into the loss function:

$$\begin{aligned} \sum_{j=0}^{T} \|z_j^{\text{fwd}} - z_j^{\text{gt}}\|_2^2 &= \sum_{j=0}^{T} \|p_j^{\text{fwd}} - p_j^{\text{gt}}\|_2^2 + \sum_{j=0}^{T} \|q_j^{\text{fwd}} - q_j^{\text{gt}}\|_2^2 \\ &\le \big(C_1^{\text{fwd}}\big)^2 \sum_{j=0}^{T} j^2 \Delta t^2 + \big(C_2^{\text{fwd}}\big)^2 \sum_{j=0}^{T} j^4 \Delta t^4 \\ &= \mathcal{O}(T^3 \Delta t^2). \end{aligned}$$

### A.2.2  REVERSAL LOSS ($\mathcal{L}_{reverse}$) ANALYSIS.

Next we analyze the reversal loss $\sum_{j=0}^{T} \|R(z_j^{\text{rev}}) - z_j^{\text{fwd}}\|_2^2$. For this, we need to refine the Taylor expansion residual terms for a more in-depth analysis.

First reconsider the forward process. Since the process is generated from the learned network, we may assume that for some constants $c_1$, $c_2$, and $c_3$, the states satisfy the following for any $0 \le j < T$:

$$\begin{cases} q_j^{\text{fwd}} = q_{j+1}^{\text{fwd}} - (p_{j+1}^{\text{fwd}}/m)\Delta t + (\dot{p}_{j+1}^{\text{fwd}}/2m)\Delta t^2 + \mathbf{rem}_j^{\text{fwd},3}, & \text{(21a)} \\ p_j^{\text{fwd}} = p_{j+1}^{\text{fwd}} - \dot{p}_{j+1}^{\text{fwd}}\Delta t + \mathbf{rem}_j^{\text{fwd},2}, & \text{(21b)} \\ \dot{p}_j^{\text{fwd}} = \dot{p}_{j+1}^{\text{fwd}} + \mathbf{rem}_j^{\text{fwd},1}, & \text{(21c)} \end{cases}$$

where the remaining terms $\left\|\mathbf{rem}_j^{\text{fwd},i}\right\|_2 \leq c_i \Delta t^i$ for $i = 1, 2, 3$. Similarly, we have approximate Taylor expansions for the reverse process:

$$
\begin{cases}
\boldsymbol{q}_j^{\text{rev}} = \boldsymbol{q}_{j+1}^{\text{rev}} + (\boldsymbol{p}_{j+1}^{\text{rev}}/m)\Delta t + (\dot{\boldsymbol{p}}_{j+1}^{\text{rev}}/2m)\Delta t^2 + \mathbf{rem}_j^{\text{rev},3}, & \text{(22a)} \\[2mm]
\boldsymbol{p}_j^{\text{rev}} = \boldsymbol{p}_{j+1}^{\text{rev}} + \dot{\boldsymbol{p}}_{j+1}^{\text{rev}}\Delta t + \mathbf{rem}_j^{\text{rev},2}, & \text{(22b)} \\[2mm]
\dot{\boldsymbol{p}}_j^{\text{rev}} = \dot{\boldsymbol{p}}_{j+1}^{\text{rev}} + \mathbf{rem}_j^{\text{rev},1}, & \text{(22c)}
\end{cases}
$$

where $\left\|\mathbf{rem}_j^{\text{rev},i}\right\|_2 \leq c_i \Delta t^i$ for $i = 1, 2, 3$.

We will prove via induction that for $k = T, T - 1, \cdots, 0$,

$$
\begin{cases}
\|R(\boldsymbol{q}_k^{\text{rev}}) - \boldsymbol{q}_k^{\text{fwd}}\|_2 \leq C_3^{\text{rev}}(T - k)^3 \Delta t^3, & \text{(23a)} \\[2mm]
\|R(\boldsymbol{p}_k^{\text{rev}}) - \boldsymbol{p}_k^{\text{fwd}}\|_2 \leq C_2^{\text{rev}}(T - k)^2 \Delta t^2, & \text{(23b)} \\[2mm]
\|R(\dot{\boldsymbol{p}}_k^{\text{rev}}) - \dot{\boldsymbol{p}}_k^{\text{fwd}}\|_2 \leq C_1^{\text{rev}}(T - k)\Delta t, & \text{(23c)}
\end{cases}
$$

where $C_1^{\text{rev}}$, $C_2^{\text{rev}}$ and $C_3^{\text{rev}}$ are constants.

The entire proof process is analogous to the previous analysis of Reconstruction Loss.

**Base Case** $k = T$: Since the reverse process is initialized by the forward process variables at $k = T$, it is obvious that $\left\|\boldsymbol{q}_T^{\text{fwd}} - \boldsymbol{q}_T^{ev}\right\|_2 = \left\|\boldsymbol{p}_T^{\text{fwd}} - \boldsymbol{p}_T^{\text{rev}}\right\|_2 = \left\|\dot{\boldsymbol{p}}_T^{\text{fwd}} - \dot{\boldsymbol{p}}_T^{\text{rev}}\right\|_2 = 0$. Thus (23a), (23b) and (23c) all hold for $k = 0$.

**Inductive Hypothesis:** Assume the inequalities (23b), (23a) and (23c) hold for $k = j + 1$, which means:

$$
\begin{cases}
\|R(\boldsymbol{q}_{j+1}^{\text{rev}}) - \boldsymbol{q}_{j+1}^{\text{fwd}}\|_2 \leq C_3^{\text{rev}}(T - (j + 1))^3 \Delta t^3, & \text{(24a)} \\[2mm]
\|R(\boldsymbol{p}_{j+1}^{\text{rev}}) - \boldsymbol{p}_{j+1}^{\text{fwd}}\|_2 \leq C_2^{\text{rev}}(T - (j + 1))^2 \Delta t^2, & \text{(24b)} \\[2mm]
\|R(\dot{\boldsymbol{p}}_{j+1}^{\text{rev}}) - \dot{\boldsymbol{p}}_{j+1}^{\text{fwd}}\|_2 \leq C_1^{\text{rev}}(T - (j + 1))\Delta t, & \text{(24c)}
\end{cases}
$$

**Inductive Proof**: We need to prove (23b) (23a) and (23c) holds for $k = j$.

First, for (23c), using (21c) and (22c), we get for any $j$ that

$$
\begin{aligned}
\left\|R(\dot{\boldsymbol{p}}_j^{\text{rev}}) - \dot{\boldsymbol{p}}_j^{\text{fwd}}\right\|_2 &= \left\|(\dot{\boldsymbol{p}}_{j+1}^{\text{rev}} + \mathbf{rem}_j^{\text{rev},1}) - (\dot{\boldsymbol{p}}_{j+1}^{\text{fwd}} + \mathbf{rem}_j^{\text{fwd},1})\right\|_2 \\
&\leq \left\|R(\dot{\boldsymbol{p}}_{j+1}^{\text{rev}}) - \dot{\boldsymbol{p}}_{j+1}^{\text{fwd}}\right\|_2 + \|\mathbf{rem}_j^{\text{rev},1}\|_2 + \|\mathbf{rem}_j^{\text{fwd},1}\|_2 \\
&\leq C_1^{\text{rev}}(T - j - 1)\Delta t + 2c_1\Delta t,
\end{aligned}
$$

where the first inequality uses the triangle inequality, and the second inequality plugs in (24c). Thus taking $C_1^{\text{rev}} = 2c_1$, the above is upped bounded by $C_1^{\text{rev}}(T - j)\Delta t$, and (23b) holds for $j$.

Second, for (24b), using (21b) and (22b), we get

$$
\begin{aligned}
\left\|R(\boldsymbol{p}_j^{\text{rev}}) - \boldsymbol{p}_j^{\text{fwd}}\right\|_2 &= \left\| -\left(\boldsymbol{p}_{j+1}^{\text{rev}} + \dot{\boldsymbol{p}}_{j+1}^{\text{rev}}\Delta t + \mathbf{rem}_j^{\text{rev},2}\right) - \left(\boldsymbol{p}_{j+1}^{\text{fwd}} - \dot{\boldsymbol{p}}_{j+1}^{\text{fwd}}\Delta t + \mathbf{rem}_j^{\text{fwd},2}\right)\right\|_2 \\
&\leq \left\|R(\boldsymbol{p}_{j+1}^{\text{rev}}) - \boldsymbol{p}_{j+1}^{\text{fwd}}\right\|_2 + \left\|R(\dot{\boldsymbol{p}}_{j+1}^{\text{rev}}) - \dot{\boldsymbol{p}}_{j+1}^{\text{fwd}}\right\|_2 \Delta t \\
&\quad + \|\mathbf{rem}_j^{\text{rev},2}\|_2 + \|\mathbf{rem}_j^{\text{fwd},2}\|_2 \\
&\leq \left[C_2^{\text{rev}}(T - j - 1)^2 + C_1^{\text{rev}}(T - j - 1) + 2c_2\right]\Delta t^2,
\end{aligned}
$$

where the first inequality uses the triangle inequality, and in the second inequality we use (24a) and (24b). Thus taking $C_2^{\text{rev}} = \max\{C_1^{\text{rev}}/2, 2c_2\}$, we have the final expression above is upper bounded by $C_2^{\text{rev}}(T - j)^2 \Delta t^2$, and so the claim holds for $j$.

Finally, for (24a), we use (21a) and (22a) to get

$$
\begin{aligned}
\left\|R(\boldsymbol{q}_j^{\text{rev}}) - \boldsymbol{q}_j^{\text{fwd}}\right\|_2 &= \left\|\left(\boldsymbol{q}_{j+1}^{\text{rev}} + (\boldsymbol{p}_{j+1}^{\text{rev}}/m)\Delta t + (\dot{\boldsymbol{p}}_{j+1}^{\text{rev}}/2m)\Delta t^2 + \mathbf{rem}_j^{\text{rev},3}\right) \right. \\
&\quad \left. - \left(\boldsymbol{q}_{j+1}^{\text{fwd}} - (\boldsymbol{p}_{j+1}^{\text{fwd}}/m)\Delta t + (\dot{\boldsymbol{p}}_{j+1}^{\text{fwd}}/2m)\Delta t^2 + \mathbf{rem}_j^{\text{fwd},3}\right)\right\|_2 \\
&\leq \left\|R(\boldsymbol{q}_{j+1}^{\text{rev}}) - \boldsymbol{q}_{j+1}^{\text{fwd}}\right\|_2 + \frac{1}{m}\left\|R(\boldsymbol{p}_{j+1}^{\text{rev}}) - \boldsymbol{p}_{j+1}^{\text{fwd}}\right\|_2 \Delta t \\
&\quad + \frac{1}{2m}\left\|R(\dot{\boldsymbol{p}}_{j+1}^{\text{rev}}) - \dot{\boldsymbol{p}}_{j+1}^{\text{fwd}}\right\|_2 \Delta t^2 + \|\mathbf{rem}_j^{\text{rev},3}\|_2 + \|\mathbf{rem}_j^{\text{fwd},3}\|_2 \\
&\leq \left[C_3^{\text{rev}}(T - j - 1)^3 + \frac{C_2^{\text{rev}}}{m}(T - j - 1)^2 + \frac{C_1^{\text{rev}}}{2m}(T - j - 1) + 2c_3\right]\Delta t^3,
\end{aligned}
$$

where the first inequality uses the triangle inequality, and in the second inequality we use (24a), (24b) and (24c). Thus taking $C_3^{\text{rev}} = \max\{C_2^{\text{rev}}/3m, C_1^{\text{rev}}/6m, 2c_3\}$, we have the final expression above is upper bounded by $C_3^{\text{rev}}(T-j)^3\Delta t^3$, and so the claim holds for $j$.

Since both the base case and the inductive step have been proven, by the principle of mathematical induction, (23b), (23a) and (23c) hold for all $k = T, T-1, \cdots, 0$.

With this we finish the proof by plugging (23b) and (23a) into the loss function:

$$\sum_{j=0}^{T}\|R(\boldsymbol{z}_j^{\text{rev}}) - \boldsymbol{z}_j^{\text{fwd}}\|_2^2 = \sum_{j=0}^{T}\|R(\boldsymbol{p}_j^{\text{rev}}) - \boldsymbol{p}_j^{\text{fwd}}\|_2^2 + \sum_{j=0}^{T}\|R(\boldsymbol{q}_j^{\text{rev}}) - \boldsymbol{q}_j^{\text{fwd}}\|_2^2$$

$$\leq \left(C_2^{\text{rev}}\right)^2 \sum_{j=0}^{T}(T-j)^4\Delta t^4 + \left(C_3^{\text{rev}}\right)^2 \sum_{j=0}^{T}(T-j)^6\Delta t^6$$

$$= \mathcal{O}(T^5\Delta t^4).$$

### A.3 ANALYSIS ON IMPLEMENTATIONS OF REVERSAL LOSS

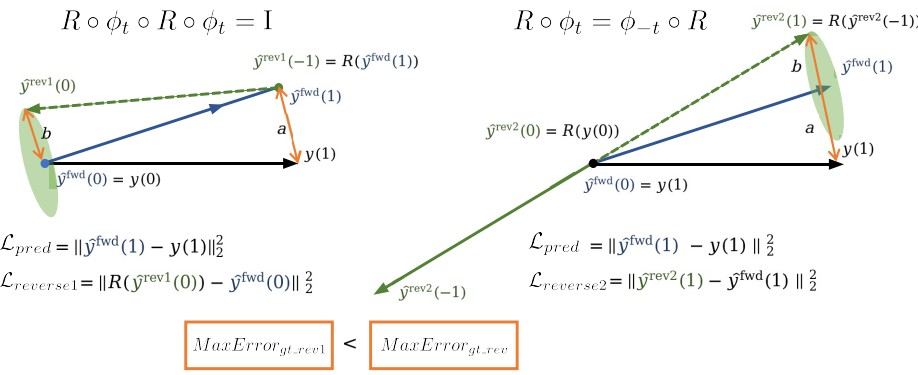

Figure 7: Comparison between two reversal loss implementation

Our time-reversal loss implementation builts upon Lemma1 where the backward trajectory originates from the last state of the forward trajectory. One could also implement the reversal loss based on Eqn 5 which is adopted in TRS-ODEN (Huh et al., 2020). We illustrated the comparison between the two implementation in Figure 7.

Here, we provide the following Lemma to show their difference: