# OpenReview forum: "TANGO: Time-Reversal Latent GraphODE for Multi-Agent Dynamical Systems"
_ICLR.cc/2024/Conference — Submitted to ICLR 2024_

### Official Review · Reviewer_zZGZ · 2023-10-23

**Soundness:** 3 good
**Presentation:** 3 good
**Contribution:** 4 excellent
**Rating:** 8
**Confidence:** 4

**Summary:**

In this paper, the authors propose a novel method with a soft constraint regularization term. The term is designed following the Time-Reversal Symmetry principle, aligns the forward and backward trajectories prediction, and thus helps the whole model to learn the dynamics of a complex multi-agent system even if it does not have the law of energy conservation. The experiments show the performance of the proposed method thoroughly on simulated datasets.

**Strengths:**

- The proposed method introduces a novel and significant contribution to the dynamics modeling field. It tackles the challenging task of modeling systems that deviate from the conventional energy conservation principles. Notably, the introduction of TANGO appears to be a fresh addition, enriching the research landscape.
- The paper exhibits a well-organized structure, systematically introducing and showcasing the performance of TANGO, making it easily understandable for readers.
- The work is thoughtfully situated within the existing literature, providing context and relevance to the broader research landscape.
- The comprehensive and compelling experiments conducted on simulated data greatly enhance the paper's credibility and the method's efficacy.
- The inclusion of well-structured and clean code is commendable, ensuring the reproducibility of the results and making it accessible for future research endeavors.

**Weaknesses:**

- The current method description appears to be somewhat confusing and would benefit from revision. For instance, in Section 3.1, Equation (6) poses challenges in understanding aspects like the derivation of $z_1^{\text{fwd}}(t)$, and it's worth noting that $g(\cdot)$ is missing in Figure 2.
- While the experiments are well-executed on simulated data, there's a notable absence of real-world data. This could potentially be attributed to the inherent challenges associated with collecting such data, as acknowledged in the paper.

**Questions:**

- In the introduction, the paragraph preceding "Contributions" mentions the "time-reversal loss" not requiring additional labels beyond "ground-truth observations." Clarification is needed regarding what constitutes these "ground-truth observations."
- It's worth exploring the adaptability of TANGO to dynamic graphs, which involve nodes and edges appearing and disappearing. This could be an intriguing avenue for future research.
- Notably, the results in Table 1 reveal that TANGO exhibits a lower Mean Squared Error (MSE) on the Damped Spring dataset compared to the Simple Spring dataset, which is somewhat surprising. Investigating the underlying reasons for this discrepancy may shed light on the method's performance characteristics and potential areas for improvement.
- A suggestion: please check the references and make them up-to-date.

---

> ### Author Response · Authors · 2023-11-21
>
> We thank you for championing our paper and your insightful comments for improving our draft. We would like to make the following claims to address your concerns.
>
> >W1: About writing and datasets.
>
> A1:  We sincerely thank reviewers for valuable suggestions. We will revise our writing and Figure 1 based on your comments for audiences to better understand our work. Regarding the application of TANGO to real-world dataset, we have applied our method to a new motion capture dataset shown below and observed improvement compared with baselines.  The reason in the submission we only utilize simulated datasets is that most real-world datasets contain more noise and the underlying dynamics are unknown to us, therefore hard to study time-reversal symmetry at the beginning. The added motion dataset is one example that how this physical-informed learning can help, and we will further investigate how TANGO can be used for other real-world applications in the future.
>
> Table 1: Evaluation results on MSE ($10^{-2}$) on MoJoCo Dataset (Huang et.al. Neurips 2020)
>
> | $\text{Model \quad }$        | $\text{LatentODE \quad}$ | $\text{HODEN \quad}$  | $\text{TRS-ODEN}_\text{GNN} \quad$ | $\text{LGODE} \quad$   | $\text{TANGO} \quad$   |
> |--------------|-----------|----------|-------------------------|---------|---------|
> | MSE          | 2.9061     | 1.9855 |0.2609                  | 0.7610  | 0.2192  |
>
>
> >Q1: In the introduction, the "time-reversal loss" does not require additional labels beyond "ground-truth observations." Clarification is needed regarding what constitutes these "ground-truth observations."
>
> A1: Thank you for bringing this up. Here, “ground-truth observations” refers to the observed training trajectories, such as particles’ positions and velocities. We want to express that in order to compute the time-reversal loss, we do not need additional data input compared with purely data-driven ones such as LG-ODE (Huang et.al. Neurips 2020). Sorry for the misunderstanding and we will clarify this in our revised revision.
>
> >Q2:It's worth exploring the adaptability of TANGO to dynamic graphs, which involve nodes and edges appearing and disappearing. This could be an intriguing avenue for future research.
>
> A2: Thanks for raising this interesting question! TANGO has the flexibility to use any ODE function for the target task. In terms of dynamic graph modeling, a simple solution is to modify the current ODE network to be CG-ODE (Huang et.al. KDD 2021) which captures the co-evolution of nodes and edges. We would like to emphasize that TANGO is a general framework to inject the inductive bias into varying dynamical systems. This is because we achieve this through a novel regularization term, and the ODE function can be of any form, depending on the target systems.
>
> >Q3: Notably, the results in Table 1 reveal that TANGO exhibits a lower Mean Squared Error (MSE) on the Damped Spring dataset compared to the Simple Spring dataset.
>
> A3: We believe the main reason behind this observation is that the velocities of the particles in the damped spring system are gradually decreasing. Therefore, the numerical scale of MSE is smaller compared to the simple spring system. We will report the MAPE (Mean Absolute Percentage Error) as complementary to MSE to enable direct comparison between different dynamic systems.
>
> [1] CMU. Carnegie-mellon motion capture database. 2003. URL http://mocap.cs.cmu.edu.

---

> > ### Comment · Reviewer_zZGZ · 2023-11-21
> >
> > Dear Authors,
> >
> > Thank you for your detailed rebuttal. However, I have a follow-up question concerning my initial review's Question 1. Upon examining the code, I noticed that the adjacency matrix is utilized within the GNN-based ODE framework. Given that the operation of a GNN inherently requires an adjacency matrix, could you clarify if this matrix represents a form of ground-truth data?
> >
> > Looking forward to your response.
> >
> > Kind regards,
> > Reviewer zZGZ

---

> > > ### Author Response · Authors · 2023-11-21
> > > **Response for Reviewer zZGZ**
> > >
> > > Thank you for your rapid response! Yes, our model input consists of both historical trajectories and the graph structure input (i.e. the adjacency matrix). The reason we need the adjacency matrix is to account for the mutual interactions among agents (both in the encoder when inferring their latent initial states and in the ODE for generating their future trajectories). In the case of evolving edges, our model input would be a sequence of graph structure inputs, and the ODE function can be designed to learn the evolution of edges as in Huang et.al. KDD 2021 [1].
> > >
> > > [1] Huang et.al. Coupled Graph ODE for Learning Interacting System Dynamics. KDD 2021.

---

> > > > ### Comment · Reviewer_zZGZ · 2023-11-22
> > > >
> > > > Dear Authors,
> > > >
> > > > Thank you for your response to my review. I must emphasize the urgency of revising the paper to clarify certain aspects that are currently misleading. For instance, in Section 2, the statement "Model input consists of trajectories of such features..." and similar sentences have led to a misunderstanding of your work. This issue of clarity regarding what exactly constitutes the model input is significant, and I believe further revision is necessary to ensure that readers have a clear and accurate understanding of your methodology.
> > > >
> > > > In light of these concerns, I find it necessary to lower my evaluation score at this time. However, I am open to revising my score upwards if the necessary amendments are made and they align with my expectations.
> > > >
> > > > Thank you once again for your attention to this matter, and I look forward to reviewing the revised version of your paper.
> > > >
> > > > Warm regards,
> > > > Reviewer zZGZ

---

> ### Author Response · Authors · 2023-11-22
>
> Thank you for your comprehensive review. We've made revisions to our manuscript, incorporating the graph input highlighted in red.
>
> At the top of Sec 2.PRELIMINARIES AND RELATED WORKS, we introduced the graph $\mathcal{G} = (\mathcal{V},\mathcal{E})$. In this paper, we assume the graph structure is given and remains static along the time.  We'd like to mention that in our 'Figure 2: Overall framework of TANGO' and  the  in function$ f_{ENC}()$ in Eqn (1)(6), we state that the input encompasses both the graph
> $\mathcal{g}$ and sequence $X(t_{1:K})$ . We do not intentionally obscure this input element. If there is anything else that remains unclear, we are more than willing to make further revisions.

---

> > ### Author Response · Authors · 2023-11-23
> >
> > Dear Reviewers,
> >
> > Thank you for your invaluable feedback. With the deadline for the author-reviewer discussion phase drawing near, we wish to ensure that our response sufficiently addressed your concerns regarding the input and the revised version of our paper. We hope this align with your expectations and positively influence the score. Should there be any outstanding issues or the need for further clarification, please do not hesitate to reach out to us.
> >
> > Best Regards,
> > Authors

---

> > > ### Comment · Reviewer_zZGZ · 2023-11-23
> > >
> > > Dear Authors,
> > >
> > > I am happy with the latest revision as it states clearly the input with $\mathcal{G}$. I raised my score.
> > >
> > > Warm regards,
> > > Reviewer zZGZ

---

### Official Review · Reviewer_qpBm · 2023-10-30

**Soundness:** 2 fair
**Presentation:** 3 good
**Contribution:** 2 fair
**Rating:** 5
**Confidence:** 3

**Summary:**

The paper introduces TANGO, a novel GraphODE model designed to model multi-agent dynamical systems by including a loss term introducing a time reversal symmetry constraint. This term aligns forward and backward trajectories of the learned GraphODE model. The authors argue that given the prevalence of time reversal symmetry in many physical systems, this introduces a useful inductive bias that can be helpful even for systems that don't adhere strictly to time reversal symmetry. They provide a theoretical treatment of the error terms introduced by this time reversal symmetry loss. They provide comparisons on 4 benchmarks between several methods for learning dynamics from irregularly sampled trajectories, several of them based on GraphODEs. The authors compare their loss in an ablation studies with 2 different implementations of a similar TRS loss.

**Strengths:**

The idea of integrating time reversal symmetry into machine learning models is interesting, and comparing different formulations with each other, and investigating to which extent these implementations differ in training stability and numerical errors, is a relevant area of study. I particularily appreciated the formal treatment of time reversal symmetry as accounting for numerical errors in Theorem 1 as an interesting contribution.

**Weaknesses:**

Main Contribution and Ablation Study

As the authors point out, closely related ideas to those one proposed here have been around for a while (see e.g. https://arxiv.org/abs/2003.02236 for Koopman operators, or, as the authors point out, in a very similar form in Huang et al. 2020).
To my understanding, the main contribution is a slight reformulation of the consistency loss from Huang et al. 2020, Eq.11, so that the consistency is computed between forward and backward generated trajectories (as a form of self-consistency) and not between the predicted forward-and backward trajectories and the ground-truth data. They argue that when performing the comparison to ground-truth data, “such implicit regularization does not force time-reversal symmetry, but introduce more noise”.
The comparison of their approach with the alternative of the time reversal symmetry loss (gt-rev in Table 1) are not as convincing to me as the authors claim, since they appear quite close to their own results. Without provided statistics, it's challenging to gauge the robustness of this effect, especially given the unexpected four-digit precision in a stochastic optimization setting.

The whole paragraph explaining the ablation study (“Finally, we conduct two ablation by changing the implementation”) I found a little confusing, also given there are several grammatical inconsistencies here. Since this reformulation is highlighted as the main contribution, it would benefit from further elaboration and clarity.

Choice of comparisons:

Comparing to single-agent models on benchmarks that seem tailored to favor the inductive bias of multi-agent systems may not be particularly enlightening. Given the contribution is about improving performance with a specific formulation of a time reversal symmetry loss, investigating this for other dynamical systems and models, and e.g. comparing performance for several types of ODE models both with and without this time reversal symmetry loss, would make more sense in my opinion.

Limited Application:

The proposal in this paper, focusing on time reversal symmetry, isn't exclusive to latent graph ODEs or multi-agent dynamical systems, and has been studied in other contexts. As the authors mention, time reversal symmetry is a universal physical principle, and its incorporation through a time-reversal-sensitive loss is potentially more far-reaching. Merely extending it to a pre-existing architecture (GraphODEs) doesn't, in itself, signify a substantial contribution. The real value of the present study to my mind lies in the exploration of different forms of this time reversal consistency loss. This exploration could be more valuable if taken beyond the confines of multi-agent GraphODEs. Discussions about its applicability to other dynamical system architectures and other, e.g. single-agent systems, would make sense in this context. An investigation into Theorem 1's implications for the loss in Eq. 5 might also be interesting, since I'd suspect the scaling with sequence length T is different here. The conclusion could also discuss the potential of this loss in various contexts.

While there are interesting first steps taken in this paper, there is still room to extend these ideas to a wider, more generally applicable framework and theoretical investigation. Otherwise the contribution is quite limited and the experimental underpinning not fully convincing. If these concerns are addressed I am happy to adjust my score.

**Questions:**

Why does the MSE of the chaotic pendulum remain almost constant across prediction lengths in Fig. 4? For a chaotic system, I would expect exponential divergence of trajectories? How does the time horizon compare to the system's approximate prediction horizon, related to its maximum Lyapunov exponent?

Figure 5:
The left plot lacks x and y-axis labels, which could aid with interpretation. On the right side, including ground truth system energies, similar to Fig. 1, would be beneficial for clarity.

---

> ### Author Response · Authors · 2023-11-21
>
> We thank you for your valuable suggestions for our work. However, we believe there’s some misunderstanding and would like to make the following claims to address your concerns.
>
> >W1: About Main Contribution and Ablation Study.
>
> A1:  We would like to emphasize our contributions  1.) Our work is the first to establish the theoretical connection between the physical meaning of the time reversal symmetry loss with its numerical benefits in improving the accuracy in general, without requiring the systems to follow any energy constraint (Theorem 1). Specifically, we show that our time reversal loss is minimizing higher-order Taylor expansion terms. Therefore, we broaden the application of the proposed loss to more real-life dynamical systems. 2.) Secondly, the consistency loss in Azencot et.al. ICML 2020 is to preserve time reversibility, instead of time-reversal symmetry in our paper. Time reversibility just requires the forward and backward trajectories to be the same, without assuming the underlying dynamics to be the same [1], i.e. the functions that generate them can be different, such as the matrix C,D in Azencot et.al. ICML 2020. In our paper, we align the forward and backward trajectories produced by the same ODE function. This time-reversal symmetry is a  fundamental physical property in many dynamical systems with definition in Eqn 5. We thank the reviewer for bringing this paper to our attention and we will discuss it in our related work and are happy to explore time reversibility in our future work.
>
> Our key motivation and novelty of the time-reversal loss design lies in the equivariant definition in Lemma 5. Time-reversal symmetry has different definitions of equivalence. The loss functions in our paper and in Huh et.al. Neurips 2020 are built upon definitions in Lemma 1 and Eqn (5) respectively, which are approximations to softly satisfy the definitions. We theoretically show in Appendix A.3 that our reversal loss is guaranteed to have a lower maximum error compared to the approximation motivated by Eqn 5 proposed by Huh et al NeurIPS 2020. This is also shown empirically in Table 1: $TANGO_{L_{rev}=\text{rev2}}$ is a model variant by replacing our reversal loss with the one proposed in Huh et al NeurIPS 2020 based on Eqn 5. TANGO consistently outperforms $TANGO_{L_{rev}=\text{rev2}}$  across systems, showing the superiority of our approximation.
>
> The second model variant $TANGO_{L_{rev}=\text{gt-rev}}$  is by computing the reversal loss $L_{reverse}$ as between model backward predictions to ground truth, in contrast with our proposed loss between model backward and forward predictions used in TANGO. In Table 1, we can see that  $TANGO_{L_{rev}=\text{gt-rev}}$ decreases the performance by 1.20%, 5.02%,3.82%, and 28.98% for the four systems respectively. We further repeated our experiments multiple rounds and provided std in the following. We observed that TANGO consistently outperforms  $TANGO_{L_{rev}=\text{gt-rev}}$ and in general has smaller stds.
>
> Table 1: Evaluation Results on MSE ($10^{-2}$) across 5 runs
> | Model| $Simple \ Spring \quad $     | $Forced\ Spring \quad    $     | $Damped  \ Spring \quad   $  | $Pendulum \qquad \quad  $  |
> |------------|-----------|-----------|-----------|-----------|
> | $TANGO$   | 1.1101$\pm$ 0.0159 | 1.4565 $\pm$0.0176 | 0.6023$\pm$0.0112 | 1.2561$\pm$0.0021 | 0.2900$\pm$    0.0701  |
> | $TANGO_{\mathcal{L}_{rev}=\text{gt-rev}}$  | 1.1113$\pm$ 0.0162| 1.5865$\pm$0.0451    | 0.6209$\pm$0.0160 | 1.6254$\pm$0.0150    |
>
> [1] Sozzi, Marco. Discrete symmetries and CP violation: From experiment to theory.2008
>
> >W2: Choice of compared baselines:
>
> A2 : Our time-reversal symmetry loss is indeed applicable to many existing neural ODE-based models for modeling dynamical systems. However, most existing neural ODE models are tailored for single-agent dynamical systems except GraphODE in Huang et.al. Neurips 2020, where the dynamical laws are usually less complicated, without complex interactions among agents. We therefore decide to validate its effectiveness in a more challenging multi-agent dynamical system setting, where the learning task is in general more complicated. For the time-reversal loss designed for single-agent systems (Huh et.al. Neurips 2020), we show it is less effective compared to ours by transferring it into the multi-agent setup (i.e., results of TRSODEN-GNN).

---

> ### Author Response · Authors · 2023-11-21
>
> >W3: Limited Application.
>
> A3: We thank the reviewer for acknowledging our theoretical contribution to the proposed time-reversal symmetry loss. We would like to claim that Theorem 1 largely broadens the application of our loss : besides its physical meaning that it can be beneficial to systems that follow time-reversal symmetry,  the loss from the numerical aspect is minimizing higher-order Taylor expansion terms, which is broadly applicable to numerous systems regardless of whether they conserve energy or obey time-reversal symmetry.  Such numerical benefit is general enough for both single-agent and multi-agent dynamical systems, thus largely broadening our model's application field. As we mentioned above, usually multi-agent dynamical systems are far more complex for a model to learn, we therefore choose to empirically validate our model’s effectiveness in the multi-agent setting across varying dynamical laws.
> To address your concern, we additionally added a real-world multi-agent system about human motion (Huang et.al. Neurips 2020). From the following table, we can see TANGO consistently outperforms existing baselines.  Thanks to your valuable advice, we will emphasize the broad applications of our model in the revised version,  and we are happy to conduct more experiments on single-agent systems and more challenging multi-agent systems in the future.
>
> Table 1: Evaluation results on MSE ($10^{-2}$) on MoJoCo Dataset (Huang et.al. Neurips 2020)
>
> | $\text{Model \quad }$        | $\text{LatentODE \quad}$ | $\text{HODEN \quad}$  | $\text{TRS-ODEN}_\text{GNN} \quad$ | $\text{LGODE} \quad$   | $\text{TANGO} \quad$   |
> |--------------|-----------|----------|-------------------------|---------|---------|
> | MSE          | 2.9061     | 1.9855 |0.2609                  | 0.7610  | 0.2192  |
>
>
> >Q1: The prediction performance for the Pendulum dataset.
>
> A1:  We thank the reviewer for raising this question. Due to the sensitivity and complexity of the Pendulum dataset, single-agent baselines (HODEN and TRSODEN) have much larger MSE values compared with TANGO and LG-ODE. For better visualization, we plot using the LOG value of MSEs instead of their absolute values. We show the absolute values in the following table.
>
> Table 3: Evaluation results on MSE ($10^{-2}$)  for varying prediction lengths across datasets
> |  Predict Length (step) $\quad$ |   Model   | 20        | 30        | 40        | 50        | 60        |
> |-----------------|-----------------------|-----------|-----------|-----------|-----------|-----------|
> | $Simple \ Spring$  | TANGO                 | 0.9295    | 0.8351    | 0.8339    | 0.9684    | 1.2483    |
> |                 | LGODE                 | 1.0710     | 0.9695    | 1.0099    | 1.2589    | 1.7474    |
> |                 | TRS-ODEN              | 0.6333    | 1.3164    | 2.1060     | 2.9184    | 3.6947    |
> |                 | HODEN                 | 0.5281    | 1.0941    | 1.7540     | 2.4196    | 3.0039    |
> | $Damped \ Spring$  | TANGO                 | 0.7755    | 0.6703    | 0.6292    | 0.6540     | 0.7409    |
> |                 | LGODE                 | 0.8429    | 0.7419    | 0.7276    | 0.8081    | 0.9929    |
> |                 | TRS-ODEN              | 0.6042    | 0.9821    | 1.2980     | 1.5566    | 1.7595    |
> |                 | HODEN                 | 1.0277    | 2.2681    | 4.0271    | 6.3051    | 8.8082    |
> | $Forced \ Spring$   | TANGO                 | 0.8823    | 0.8232    | 0.8609    | 1.0031    | 1.2698    |
> |                 | LGODE                 | 1.1222    | 1.004     | 1.0702    | 1.3244    | 1.8012    |
> |                 | TRS-ODEN              | 1.1204    | 1.7748    | 2.5962    | 3.5295    | 4.4498    |
> |                 | HODEN                 | 0.9713    | 1.5606    | 2.3225    | 3.1997    | 4.0668    |
> | $Pendulum $       | TANGO                 | 0.9309    | 0.9723    | 1.0039    | 1.0858    | 1.2823    |
> |                 | LGODE                 | 1.3601    | 1.3446    | 1.3464    | 1.4336    | 1.5718    |
> |                 | TRS-ODEN              | 281.4785  | 359.0132  | 418.5017  | 540.5373  | 741.4988  |
> |                 | HODEN                 | 281.1073  | 358.0864  | 417.7614  | 540.9593  | 741.2296  |
>
> We also computed the Maximum Error Exponent of each model on the $Simple Spring$ and $Pendulum$ datasets, and the values are presented in the following table.
>
> Table 4: Maximum Error Exponent of each model on $Simple Spring$ and $Pendulum$.
> | Model | $Simple \ Spring$ | $Pendulum$ |
> |---------|-------------|-------------|
> | TANGO | 18.6959 | 85.7173|
> | LGODE | 22.6326| 113.5891 |
> | TRS-ODEN| 41.6580 | 821.3406 |
> | HODEN | 31.9064 | 819.0590 |
>
> The tabulated data clearly demonstrates a notably higher overall maximum error exponent for the pendulum in comparison to the spring system, indicating the pendulum system's greater chaotic nature. Moreover, among all models, TANGO consistently exhibits smallest MLE, implying its superior stability.

---

> > ### Comment · Reviewer_qpBm · 2023-11-22
> >
> > Thanks for the feedback and the novel experimental results!
> >
> > My main concern was that when selecting baselines in a way that are clearly multi-agent dynamical systems, the comparison method with single-agent systems that do not accomodate this strong inductive bias are unsurprisingly worse. This was also my issue with respect to the framing of the whole paper: the title implies that you are investigating "time-reversal graphODEs for multi-agent dynamical systems", while you here again make the point that your main contributions are actually in the numerical and theoretical investigation of a time-reversal loss that is both independent of graphODEs and independent of multi-agent dynamical systems.
> >
> > I have two follow up questions:
> > With respect to your theoretical results, I was wondering if the dependence of the loss to O(T^5) is necessarily a good thing, or can also lead to instability during training, and whether you have observed something in this direction? For chaotic systems (which your MLE imply all of your benchmarks are), you necessarily run into exploding gradient problems for longer input sequences.
> >
> > I also have a follow up question regarding the computation of the maximum Lyapunov exponents:
> > your results indicate that both the simple spring and the pendulum system are markedly in the chaotic regime (as a comparison, the Lorenz attractor has a maximum Lyapunov exponent around 0.9). I would also disagree with the statement that it is necessarily good that "TANGO consistently exhibits smallest MLE, implying its superior stability", because the MLE of the inferred system should reflect that of the ground-truth/simulated benchmark.

---

> ### Author Response · Authors · 2023-11-22
>
> We thank you for your valuable comments and suggestions. We would like to make the following claims and hope they can address your concerns.
>
>
> >Q1: About baselines selection
>
> A1: We study multi-agent system dynamics in this paper due to its more complex nature, and agree that our reversal loss can be applied to broader applications. However, this work's main focus is to demonstrate that existing pure data-driven approaches for multi-agent systems have fundamental limitations and the right direction is to integrate physics deeper into the data-driven models. We request that the fact our invention that can benefit broader applications will not be evaluated as a negative factor. We will put a discussion session on its implications to reflect the insights pointed out by the reviewer.
>
> For the baselines, we have multi-agent baselines that are purely data-driven (LGODE)  and physics-informed $TRSODEN_{GNN}$. The latter one is a valid multi-agent baseline as the inductive bias we impose on the multi-agent setting for both our method and this baseline is via GNN, which captures agents' interactions. The reason we also select single-agent baselines is that 1.) there are not many physics-informed NN that can be easily transferred to the multi-agent setting as $TRSODEN_{GNN}$. To validate whether our physics-informed design is reasonable, we want to compare it with baselines that have injected physical inductive bias in a different way such as HODEN by imposing energy conservation. 2.) It is also widely adopted in other multi-agent papers to compare against single-agent baselines [1,2] so as to justify whether the interaction among agents is important compared with the physical law we want to inject into our model.
>
> [1] Huang et.al. Learning Continuous System Dynamics from Irregularly-Sampled Partial Observations. Neurips 2020
> [2] Kipf et.al. Neural Relational Inference for Interacting Systems. ICML 2018
>
>
> >Q2: About the dependence of the loss to O(T^5)
>
> A2:  As denmonstrated in Theorem1 and the proof, the reconstruction loss $\mathcal{L}_{pred}$ is $\mathcal{O} (T^3 \Delta t^2) $.
>
> The time-reversal loss $\mathcal{L}_{reverse}$ is $\mathcal{O} (T^5 \Delta t^4) $.
>
> Regarding the relationship to $T$, $\mathcal{L}_{reverse}$ is more sensitive to total sequence length ($T^5$), thus it provides more regularization for long-context prediction, which means our models can offer greater physical guidance when facing the key challenge for dynamic modeling. And we did not encounter the issue of instability during our training process.

---

> ### Author Response · Authors · 2023-11-22
>
> >Q3: About MLE
>
> We are sorry that there might be some misunderstanding regarding the previous MLE issue.
>
> Based on my current findings, when assessing the chaos level of a system using the Maximum Lyapunov Exponent (MLE), we should use two initial states that are very close (e.g., 1.000 and 1.001), allowing them to evolve  following the systems’ dynamic for a sufficiently long time (t -> infinity). The MLE is then calculated using the formula:
>
> $\lambda=max_{t\rightarrow \inf}(\frac{1}{t}\ln \frac{||\delta(t)||}{||\delta(0)||}).(1)$
>
> We set fixed initial values for each dataset and generate 10 trajectories by perturbing the initial values with random noise (0, 0.0001). We calculate the Maximum Lyapunov Exponent (MLE) between any two trajectories. Finally, we compute the average and std of MLE from all pairs to gauge the chaotic behavior of each dataset. The data is presented in the table below:
>
>  Table 5: Maximum Lyapunov Exponent of different datasets
>
> | Dataset | $Simple \ Spring$ | $Forced \ Spring$ | $Damped \ Spring$ | $Pendulum $ |
> | --- | --- | --- | --- | --- |
> | MLE（in 60 steps） | 0.4031 $\pm$ 0.3944 | 1.0087$\pm$ 1.0577 | 0.6307 $\pm$ 0.7065 | 34.1832 $\pm$ 30.1846 |
>
> From the table, it's evident that the order of MLE values is: $Pendulum$ >> three $Spring$ datasets. This observation is consistent with the evaluation results based on MSE presented in our previous responses in Table 3 which indicates that as the prediction length(Steps*step size) increases, there is a more significant performance degradation of all models on $Pendulum$  dataset.
>
> In our previous response, in order to compare all the models shown in the figure.4 (TANGO, LGODE, HODEN, TES-ODEN) when varying prediction lengths,  we use the predicted trajectory and the ground truth trajectory instead of two trajectory with very close initial states to calculate the Eq(1).
>
> Thus on spring and pendulum datasets, we calculate four values for each model. These values are not following the precise original definition of MLE evaluating the chaos level of a system. So it is not suitable to directly compared our values with the Lorenz attractor’s 0.9.
>
> We can consider it as a metric called maximum error exponent that borrows the conceptual framework of MLE, capable of assessing whether the predictive performance of a method significantly deteriorates with increasing steps within a finite horizon. So the absolute values are not that important, and we should focus on the comparison between different models.
>
> Therefore, as depicted in Table 4 (our previous response box), given TANGO's consistent demonstration of the smallest values, we intend to convey that TANGO maintains superior prediction performance as the number of prediction steps increases. This aligns with the conclusion of our paper's Figure 4, which explores the evaluation across various prediction lengths.
>
> Moreover, for every model, the values on the pendulum dataset are larger than those on the simple spring dataset, indicating that the pendulum dataset is indeed a more complex system. This aligns with the Evaluation results on mse presented in our previous responses in Table 3.
>
> We will clarify the definition of this new metric maximum error exponent when updating our revision.

---

> > ### Comment · Reviewer_qpBm · 2023-11-23
> >
> > Thank you for the clarification. Regarding the maximum Lyapunov exponents: yes, I agree with the definition, and there are well-established methods for approximating MLE directly from data (see e.g. Rosenstein 1993).
> > Given you compute it for a discretized version of your sampled trajectories with a step size delta t, the MLE is in reciprocal units, so proportional to 1/t.
> >
> > However in practice, if the delta t you choose is very small, then the effect of the chaos is weaker on the trajectories when you compute predictions. That is why I was wondering about how the values in the previous table were normalized (what is the delta t for each step in Table 3). In the appendix you only mention "we integrate with a fixed step size and sub-
> > sample every 100 steps", but what the fixed step size is was not made explicit as far as I could see.
> >
> > This question and my previous one about stability of the time reversal loss during training were also related to recent work (see Mikhaeil, Monfared, NIPS 2022) that showed exploding error gradients to the system's underlying maximum Lyapunov exponent, and are unavoidable during training.
> >
> > Regarding baseline selection: it is true that the papers you cite select single-agent setups as comparisons. However, this is still different to my mind since these papers' main contributions are actually about novel ways of learning multi-agent systems, while as I have argued before (and you have pointed out yourself) your main theoretical and practical contributions are about the time-reversal loss and its implications and completely independent of multi-agent systems. I don't want to imply that the broader applications are a negative factor, but indeed they could be a strength of the paper. My concern is more that the specifity of the applications that you focus on in your paper, starting from the framing in the title, renders parts of your paper irrelevant in underscoring your actual contributions, while at the same time not exploring the broader applications in too much depth.
> >
> > I will raise my score thanks to your feedback and efforts during the revision, but am still a little torn between a clear recommendation for acceptance since I think the paper (and the potentially important insights within it) could benefit from a more thourough reframing.

---

> > > ### Author Response · Authors · 2023-11-23
> > >
> > > Dear reviewers,
> > >
> > > We sincerely appreciate your valuable feedback. As the deadline for the author-reviewer discussion phase is approaching, we would like to check if our response addressed your concerns regarding the step size and baseline selection. If there are any remaining issues or if you require further clarification, please feel free to inform us.
> > >
> > > Best Regards, Authors

---

> ### Author Response · Authors · 2023-11-23
>
> Thanks for your time and valuable suggestions for our work.
> >Q1: About the step size
>
> A1: The $\delta t$ for the three spring systems was set at 0.001,  and we conducted simulations involving 6000 forward $\delta t$ steps, sampling data every 100 steps. We chose the same settings following LGODE[1] and NRI [2]. From the visualization in Fig. 5, we can observe that the trajectories can reflect the intricate behaviors of multi-agent dynamical systems. We are contemplating exploring longer prediction lengths in our future work.
>
> For the pendulum dataset, due to an anticipated higher chaotic level, we opted for a smaller $\delta t$ of 0.0001, and conducted simulations involving 6000 forward $\delta t$ steps, sampling data every 100 steps.
>
>
>
> [1] Zijie Huang, Yizhou Sun, and Wei Wang. Learning continuous system dynamics from irregularlysampled partial observations.
> In Advances in Neural Information Processing Systems, 2020.
>
> [2] Thomas Kipf, Ethan Fetaya, Kuan-Chieh Wang, Max Welling, and Richard Zemel. Neural relational inference for interacting systems. arXiv preprint arXiv:1802.04687, 2018.
>
> >Q2: About baseline selection
>
> A2: We thank the reviewer for acknowledging the potential broader applications of our model. In this paper, the reason we discuss our model under the scope of multi-agent dynamical systems is two-fold: 1.) Firstly, multi-agent dynamical system modeling itself already has many important downstream applications. Our method is actually motivated by the energy explosion phenomenon (Figure 1) caused by an existing multi-agent baseline (LGODE) which is purely data-driven, and we hope our physics-informed GraphODE can overcome it. 2.) Meanwhile, we emphasize the universal applicability of the loss function so researchers can easily adapt it to systems of their interest. This actually serves as an additional benefit of our model that goes beyond the multi-agent setting, making it even more powerful in terms of application fields. We hope this addition outside of the multi-agent setting will not become a negative factor in our paper. We will take the reviewer's advice and emphasize the latter part more in our revised version.

---

### Official Review · Reviewer_g9nP · 2023-10-30

**Soundness:** 2 fair
**Presentation:** 2 fair
**Contribution:** 2 fair
**Rating:** 5
**Confidence:** 3

**Summary:**

This paper introduces TAGNO (Time-Reversal Latent GraphODE for Multi-Agent Dynamical Systems), a physics-based graph neural ODE-based model designed for learning and predicting dynamical systems. To enhance the regularization of graph ODEs, TANGO incorporates the concept of time-reversal symmetry, a well-established symmetry principle in classical mechanics. By formalizing a regularizer based on time-reversal symmetry (or, potentially, time-reversibility/invertibility), TANGO demonstrates superior learning and generalization capabilities when compared to other competing models. The authors also derive some theoretical properties, such as error bounds concerning the time step, further reinforcing the strengths of TANGO. To validate their proposed approach, the authors present empirical results involving four synthetic examples.

**Strengths:**

- TANGO, the model proposed in this study, consistently outperforms major competitors, including baseline Latent NODEs, Hamiltonian NODEs, TRS (Time-Reversal-Symmetric) NODEs, and graph NODEs, across a range of physics and dynamics forecasting problems.

- The paper is generally well-written and easy to follow (although there are some technically confusing points, please see the Weaknesses). Figure 1 and Figure 2 effectively summarize the motivation and core concept of this work.

**Weaknesses:**

**1. Reversing operator**

First, I would appreciate clarification on the precise calculation of the reversing operator $R$ for TANGO. While the reversing operator holds a significant role in addressing time-reversal symmetry, the current version of the paper lacks a detailed description and computation method for it (if this information is located in the appendix, I apologize; however, I would recommend including it in the main text of the revised version for clarity). Considering that the reversing operator is typically defined in the phase space $(q, p)$, as mentioned in footnote 3, it remains unclear how it can be computed within the latent space $z$. At first glance, I think there are possible approaches, although they are all have some concerns:

The first approach involves splitting $z$ into two components, $(z_0, z_1)$, with $z_0$ serving as a pseudo $q$ and $z_1$ as a pseudo $p$, defining $R: (z_0, z_1, t) \to (z_0, -z_1, -t)$. However, questions arise regarding the effectiveness of such a straightforward separation, as constraints similar to those found in variational Hamiltonian Monte Carlo literature might need to be imposed on $z_1$.

Secondly, one might consider reversing the observations $y = (q, p)$ directly, i.e., $\hat{R}: (q, p, t) \to (q, -p, -t)$, and then assume $R \circ z = f_{dec}(\hat{R} \circ y)$. However, in this case, the reversing operation $R$ within the latent dynamics might not be an involution.

Lastly, a simple identity operation, i.e., $z$, where $R: (z, t) \to (z, -t)$ might be employed. Currently, it appears that this operation is used in the paper. However, assuming the use of $R: (z, t) \to (z, -t)$ raises a new question, which I will discuss in the following section.

**2. Time-reversal symmetry or time-reversibility?**

Since there is no explicit definition of the reversing operator, I will assume that it solely reverses the dynamics in the context of the time direction, as described in equation (8) (i.e., solving the latent ODE in $-t$ direction). In this case, I have reservations about whether equation (9), as implemented in the paper, indeed enforces time-reversal symmetry regularization for the learned dynamics. It appears to me that equation (9) may promote time-reversibility (invertibility) rather than time-reversal symmetry. I believe it is crucial to rigorously differentiate between these two concepts within this work.

Time-reversal symmetry implies that forward and backward time dynamics are indistinguishable. In contrast, time reversibility implies that these two dynamics can be distinguished but are one-to-one mappings of each other (i.e., the dynamics have a unique solution over a given time interval). For instance, an ideal oscillator would remain indistinguishable whether it evolves with forward or reverse time dynamics, thereby clearly exhibiting time-reversal symmetry. Conversely, a damped oscillator can be distinguished in this context. The forward time evolution reduces amplitude over time, while the backward time evolution increases it. This system lacks time-reversal symmetry but still poses time-reversibility (invertibility) because when a video of a damped oscillator is played in reverse, it returns to the initial frame due to the existence of a unique solution, i.e., well-defined dynamics.

Taking this into account, equation (9) appears to be a loss function designed to promote time reversibility, as evident from the lower middle figure in Figure 2. It is worth noting that many deterministic ODEs, including those governing physical systems, are theoretically guaranteed to exhibit invertibility (uniqueness of solutions) according to the Picard–Lindelöf theorem. However, in practice, the actual solutions computed by numerical solvers like RK45 may not always be perfectly invertible, especially in the case of chaotic systems and homoclinic orbits. It is conceivable that the proposed regularization of TANGO compensates for such deviations from the limited precision of numerical solvers or sensitivity on initial conditions for chaotic systems (does this align with the findings of Theorem 1? I haven't verified the appendix). Alternatively, TANGO might learn smoother and more noise-resistant dynamics thanks to matching the forward and inverse trajectories.

I do not intend to argue that TANGO lacks technical validity; it is reasonable to expect that TANGO's predictive performance can be improved by introducing the proposed regularization, regardless of whether the target dynamics are conservative, non-conservative, T-symmetric, or T-asymmetric. However, I find it challenging to assert that the added regularization arises from time-reversal symmetry based on the current presentation of the paper. Therefore, I remain somewhat unconvinced by the argument presented in this paper.

**Questions:**

Please see the Weaknesses also.

- What is the precise definition of the reversing operator employed in TANGO? More broadly, is there a standardized or canonical approach to defining the reversing operator for latent-based dynamics?

- Does the suggested reversal loss function result in time-reversal symmetry? Alternatively, does it promote time-reversibility (= invertibility, solution uniqueness, reduction of numerical divergence, ...), as previously mentioned?"

- It would be valuable to explore the influence of the solver choice, particularly for LGODE, TANGO, TANGO (gt-rev), and TANGO (rev2) models, through ablation studies. For instance, what would be the outcome when employing a higher-order solver in lieu of the conventional RK45? Or, how would lower-order solvers like Euler impact the results? How about Leapfrog? Assuming that the proposed loss function indeed promotes time-reversibility (invertiblity), it is plausible that the effectiveness of TANGO could diminish as the solver's precision increases.

- In Figure 5, it is noticeable that HODEN, a model known for strict energy conservation, exhibits energy divergence in the damped spring system. What could be the reason for this occurrence?

- Some minor typos, for example, page 3: real *worldsystems*, page 4: $\tau_1,\tau_2 \in \mathbb{R}^\textit{4}$, Theorem 1: The *reversal* loss $\mathcal{L}_{pred}$, ...

---

> ### Author Response · Authors · 2023-11-21
> **Response to Reviewer g9nP (Part I)**
>
> We thank you for your valuable comments and suggestions for improving our paper. We would like to make the following claims and hope they can address your concerns.
>
> >W1: The reversing operator in the latent space and its connection to time-reversal symmetry.
>
> A1:  The implementation of the time-reversal symmetry in the latent space is based on $\hat{R}: (z,t) -> (z,-t)$, where we follow Huang, et.al. Neurips 2020, Huh et al NeurIPS 2020 in that we do not separately model q,p. This implementation is to approximate time-reversal symmetry instead of time reversibility. This is because we generate the forward and backward trajectories using the same ODE function, and then utilize the loss to make them close to each other. It differs from time reversibility where underlying dynamics (i.e. the ODE function) of forward and backward trajectories can be different, as long as the resulting trajectories are close to each other [1].
>
> >Q1: Regarding performance when using different solvers.
>
> A1: We appreciate the reviewer acknowledging the potential of our method of compensating numerical errors brought by ODE solvers (highlighted by Theorem 1). We add the following experiments to show the performance differences when using the Euler solver compared with RK4. As shown in the following table, TANGO achieves higher improvement when using RK4 compared with Euler, indicating that it adequately minimizes the higher-order Taylor expansion term to decrease numerical errors. We would like to mention that Leapfrog is not a suitable solver in our setting. As our motivation is to softly satisfy time-reversal symmetry so that it can be applicable to various dynamical systems. Leapfrog strictly enforces time-reversal symmetry. It can harm the performance of systems that do not strictly obey time-reversal symmetry such as damped/forced springs shown in Table 1.
>
> Table 1: Evaluation results on MSE ($10^{-2}$) on Different Solvers.
>
> |           | $Simple$|$Spring\qquad $   | $Forced$|$Spring\qquad $       | $Damped$ |$Spring\qquad $        | $Pendulum $ | $\qquad $      |
> |-----------|-------------------|---------|-------------------|-------|-------------------|-------|-------------|-------|
> |           | Euler             | RK4   | Euler             | RK4   | Euler             | RK4   | Euler       | RK4   |
> | LGODE     | 1.8443            | 1.7429| 2.0462            | 1.8929| 1.1686            | 0.9718| 1.4634      | 1.4156|
> | TANGO     | 1.4864            | 1.1178| 1.6058            | 1.4525| 0.8070            | 0.5944| 1.3093      | 1.2527|
> | improve % | 19.4057           | 35.8655| 21.5228           | 23.2659| 30.9430           | 38.8352| 10.5303     | 11.5075|
>
> [1] Sozzi, Marco. Discrete symmetries and CP violation: From experiment to theory.2008

---

> ### Author Response · Authors · 2023-11-21
> **Response to Reviewer g9nP (Part II)**
>
> >Q2: Why HamitonianNN fails?
>
> A2: HamitonianNN is designed to preserve energies for each single agent. In multi-agent dynamical systems, the energy is only conserved as a whole, instead of each individual’s energy being conserved. Therefore, it has inferior performance.
>
> >Q3: Typos.
>
> A3: We sincerely thank the reviewer for reading our paper carefully and we will correct them in our revised version.

---

> ### Comment · Reviewer_g9nP · 2023-11-22
>
> I appreciate the authors’ thorough response and clarification to my concerns. Below is my follow-up for your response.
>
> **Regarding W1**
>
> Thank you for your comment. However, I am still not fully convinced by your explanation. Let me explain as follows.
>
> If I understand correctly, an ODE system, $dz(t)/dt = f(z(t))$, $t \in \mathbb{R}$, $z \in \mathcal{Z}$, $f: \mathcal{Z} \to T\mathcal{Z}$, is time-reversal symmetric when the following also holds:
>
>  $dR(z(t))/dt = - f(R(z(t)))$,
>
> where the minus sign indicates the time reversing and $R: \mathcal{Z} \to \mathcal{Z}$ is a reversing operator [1]. This is equal to the equation (4) of the authors' paper. Similarly, using the concept of the evolution operator $\phi_t: z(\tau) \mapsto z(\tau + t)$, the above is equivalent to
>
> $R \circ \phi_t = \phi_{-t} \circ R \Rightarrow R \circ \phi_t  \circ R \circ \phi_t = I$,
>
> which is the result of the Lemma 1 of the paper. Now, if $R$ is an identity operation ($z \to z$), as the authors responded, it means that the time-reversal symmetric dynamics satisfies
>
>  $dz(t)/dt = f(z(t))  = - f(z(t))$ $\Rightarrow  f(z) =  - f(z)$, (1)
>
> and similarly,
>
> $\phi_t = \phi_{-t} \Rightarrow \phi_t  \circ \phi_t = \phi_{2t} = I$, (2)
>
> which seem to be not very desirable: both (1) and (2) mean that if $R$ is an identity operation, then the dynamics should be stationary ($z(t) = C$) for satisfying the time-reversal symmetry. From this statement, my follow-up questions are as follows:
>
> 1. Is it meaningful to satisfy the time-reversal symmetry when the reversing operator is an identity?
>
> 2. In the paper, the authors mention that their proposed reversal loss (8 - 9) is based on the definition of Lemma 1, $R \circ \phi_t  \circ R \circ \phi_t = I$ (and if $R$ is an identity, then $\phi_t  \circ \phi_t = I$). Can the authors derive (8-9) from the Lemma step-by-step? Currently, it seems that the reversal loss (8 - 9) is rather based on $\phi_{-t}  \circ \phi_t = I$.
>
> **Regarding Q1**
>
> Thank you for your additional experiments. I was a bit surprised that the effect of the proposed loss seemed to increase with the use of a high-order solver. However, the author's explanation that the higher-order Taylor expansion term adjusts effectively by the proposed loss is convincing. I am satisfied with the authors' response.
>
> **Regarding Q2**
>
> Thank you for clarifying that for me.

---

> ### Author Response · Authors · 2023-11-22
>
> > Q1:Is it meaningful to satisfy the time-reversal symmetry when the reversing operator is an identity
> >
>
> A1: Thanks for your suggestions. However, we would like to clarify that **R is not an identity**. In our response,  the reversing operator in the latent space is defined as $\hat{R}: (z,t) \rightarrow (z,-t)$, i.e. $R(z(t))= z(-t) = z(T-t)$, where T denotes the total trajectory length. It is **not an identity transformation** $z\rightarrow z$, as z is a time-dependent variable and the reversing operator R would reverse the time index t with a negative sign.
>
> The governing differential equations of the system are as follows:
>
> $\frac{dz(t)}{dt}=f(z(t))$
>
> $\frac{dR(z(t))}{dt}=-f(R(z(t))) \rightarrow \frac{dz(-t)}{dt}=-f(z(-t))$
>
> Now let's analyze the definition of time-reversal symmetry: $R \circ \phi_t = \phi_{-t} \circ R$.
>
> For the left side: $R \circ \phi_t (z(\tau)) = R(z(\tau + t)) = z(-\tau - t)$. (1)
>
> For the right side: $\phi_{-t} \circ R(z(\tau)) = \phi_{-t} \circ (z(-\tau)) = z(-\tau - t)$. (2)
>
> It's evident that the left side is equal to the right side.
>
> Furthermore, considering $R \circ R(z(\tau))=R(z(-\tau))=z(\tau)$, thus $R \circ R=I$. Therefore, Lemma 1 remains valid, as demonstrated below:
>
> $R \circ \phi_t = \phi_{-t} \circ R$. （3）
>
> $R \circ \phi_t \circ R= \phi_{-t} \circ R \circ R$.
>
> $R \circ \phi_t \circ R = \phi_{-t} $.
>
> $R \circ \phi_t \circ R \circ \phi_t= \phi_{-t} \circ \phi_t = I$.
>
> Since the reversing operator  $\hat{R}: (z,t) \rightarrow (z,-t)$ is not an identity transformation, directly eliminating R from both sides of the Eq.(3) to get $\phi_{t} = \phi_{-t}$  is not applicable.
>
> Thus, it is meaningful to satisfy the time-reversal symmetry based on  lemma1, where the reversing operator $\hat{R}: (z,t) \rightarrow (z,-t)$ is not an identity transformation.
>
> > Q2: How the reversal loss (8 - 9) is derived from (and related to) Lemma 1.
> >
>
> A2: The definition of time-reversal symmetry is $R \circ \phi_t = \phi_{-t} \circ R$. As previously demonstrated, we have proved that $\hat{R}: (z,t) \rightarrow (z,-t)$ satisfies Lemma 1. Hence, the definition of time-reversal symmetry $R \circ \phi_t = \phi_{-t} \circ R$ is equivalent to $R \circ \phi_t \circ R \circ \phi_t = I$. This implies that if we move t steps forward, then turn backward and move for t more steps, it shall restore back to the same state.
> And it's important to note that $\hat{R}: (z,t) \rightarrow (z,-t)$ **is not an identity transformation**, so we can not directly eliminating R to get $\phi_{t} \circ \phi_{-t} =I $.
>
> In Eq. (9), $\hat{y_i}^{fwd}(\cdot)$ represents the forward steps, while $\hat{y_i}^{rev}(\cdot)$ represents the steps taken during the reverse trajectory, and we expect they should be the same, which indeed adheres to the equivalent definition in Lemma 1.
>
> Moreover, note that $\hat{y_i}^{fwd}(\cdot)$ is not equal to $\hat{y_i}^{rev}(\cdot)$, i.e. $\hat{y_i}^{fwd}(t) \neq \hat{y_i}^{rev}(t)$. They are **two different notations**. We use them as distinct markers to denote the forward and reverse trajectories. Thus, Eq. (9) does not appear as the form of $||y(t) - y(T-t)|| $, indicating that the implementation is not derived from $\phi_t \circ \phi_{-t} = I$. You can further refer to Figure 7 in the paper's appendix. The left segment of the figure details the implementation of our reversal loss.
>
> Thank you again for your valuable suggestions and your time in reviewing our paper. We hope that your concerns are adequately addressed and may consider raising the score.

---

> ### Comment · Reviewer_g9nP · 2023-11-23
>
> Thank you for your response. Here are some confusing points that I would like to mention:
>
> ***
> First, I am not sure how can $R(z(t)) = z(-t) = z(T-t)$, except for the case when $z(t)$ is a periodic function with a periodicity of $T$. Therefore, I will consider the definition of the reversing operator used in this paper to be $R(z(t)) = z(-t)$, which authors have predominantly employed in their response.
>
> ***
> Second, I know that *time-reversal symmetry is not a generic feature inherent in ODEs* overall, as depicted in Figure 1 (b.1) of the paper. For example, an ideal oscillator exhibits time-reversal symmetry, while a damped oscillator does not. Namely, time-reversal symmetry is a certain symmetry property that applicable to specific physical dynamics, rather than a general property of ODEs. Clearly, arbitrary neural ODEs would not exhibit such symmetry naturally, thus, incorporating it as a regularization might be promising when learning certain physical systems.
>
> However, my concern lies in the assertion that the property the authors label as time-reversal symmetry, i.e., $R \circ \phi_{t} \circ R \circ \phi_{t} = I$ with $R(z(t)) = z(-t)$, appears to be in fact *the reversibility (invertibility), which holds for many well-defined ODEs naturally (including the majority of neural ODEs)*, rather than indeed time-reversal symmetry.
>
> It can be simply shown by using the fact that $R \circ \phi_{t} \circ R = \phi_{-t}$ holds very generally if $R(z(t)) = z(-t)$, *whether $\phi_{t}$ is time-reversal symmetric or not*. By using $R(z(t)) = z(-t)$, one can easily check that
>
> $(R \circ \phi_{t} \circ R)(z(\tau)) = (R \circ \phi_{t}) (z(-\tau)) = R (z(-\tau + t)) = z(\tau - t) = \phi_{-t}(z(\tau))$ (a)
>
> holds *without any prior assumptions or properties on* $\phi_t$. Therefore, in fact, the Lemma 1,
>
> $R \circ \phi_{t} \circ R \circ \phi_{t} = \phi_{-t} \circ \phi_{t} = I$ (b)
>
> holds *whether the system is time-reversal symmetric or not*. It is sufficient for the time evolution to be invertible, i.e., $\phi_{-t} \circ \phi_{t} = I$. For example, the damped oscillator is not time-reversal symmetric, but still satisfies the above relation: if a damped oscillator is forward-evolved by a time $t$ and then backward-evolved by a time $-t$, it returns to its initial state (up to the numerical error of the used ODE solver). Similarly, if an arbitrary neural ODE is forward-evolved by a time $t$ and then backward-evolved by a time $-t$, it returns to its initial state also.
>
> Note that (a) and (b) do not hold generally when using classical reversing operators like $R(q, p) = (q, -p)$ [1]. In this case, regularizing (b) surely encourage the trained dynamics to be time-reversal symmetric (i.e., the hypothesis space of the trained dynamics is reduced to a "time-reversal-symmetric" domain). However, from the current definition of the reversing operator $R(z(t)) = z(-t)$, I believe that (b) encompass the reversability, rather than time-reversal symmetry. I guess that regularizing (b) with $R(z(t)) = z(-t)$ is likely to help reduce numerical errors that arise during the forward/backward evolution process, rather than encouraging the neural ODE to be time-reversal symmetric.
>
> In summary, for the acceptance of this paper, I believe that the authors should provide a more rigorous distinction between time-reversal symmetry and time reversibility (invertibility), and they need to enhance their logic. However, based on the current version of the paper and the authors' response, I still find it challenging to definitively ascertain whether the outcomes presented by the authors conform to the concept of time-reversal symmetry. As a result, I would like to maintain my current score.
>
> ***
>
> [1] Lamb, J. S., & Roberts, J. A. (1998). Time-reversal symmetry in dynamical systems: a survey. Physica D: Nonlinear Phenomena, 112(1-2), 1-39.

---

> ### Author Response · Authors · 2023-11-23
>
> >Q1: About R, $\phi_t$ ,reversibility and time reversal symmetry.
>
> Firstly, we would like to restate the definition of Time-Reversal Symmetry in our paper
>
> “The system is said to follow the Time-Reversal Symmetry if it satisfies
> $\frac{dR(\boldsymbol{Z}(t))}{dt}=-F(R(\boldsymbol{Z}(t)),$
> where $R(\cdot)$ is a reversing operator”
>
> Let's examine the state of two systems and their underlying dynamics when we apply $R \circ \phi_t \circ R \circ \phi_t = I$. One of these systems satisfies time reversal symmetry, while the other does not.
>
> **the system that satisfies Time reversal symmetry**
>
> Step1 $\phi_t:$
>
> $z(\tau)→z(\tau+t)=z(\tau)+\int_\tau^{\tau+t} F(z(s))ds$
>
> the dynamic is $dz/dt=F(z)$
>
> Step 2 $R$:
>
> $z(\tau+t)\rightarrow R(z(\tau+t))=z(-\tau-t)$
>
> Due to the definition of a system that satisfies Time reversal symmetry as show above, the dynamic is $dR(z)/dt=-F(R(z))$
>
> $\Rightarrow dz(-t)/dt=-F(z(-t))$ (1)
>
>  $\Rightarrow dz(t)/d(-t)=-F(z(t))$
>  $\Rightarrow dz(t)/dt=F(z(t))$
> Step3 $ \phi_t:$
>
> $z(-\tau-t)→z(-\tau)=z(-\tau-t)-\int_{-\tau-t}^{-\tau} F(R(z(s)))ds=z(-\tau-t)+\int_{-\tau-t}^{-\tau} F(z(s))ds$.           -
>
>  the dynamic remains $dR(z)/dt=-F(R(z))$
>
> Step 4 $R$:
>
> $R:z(-\tau)\rightarrow z(\tau)$
>
> $dR(R(z))/dt=-(-F(R(R(z))))$ ($R$ is an involution operator, so $R \circ R=I$)
>
> $\Rightarrow dz/dt=F(z)$
>
> Thus we can see that both the state and the underlying dynamic were the same as the initial state.
>
> However, for previous pure-data driven neural networks that **don't adhere to Time Reversal Symmetry**, Eq(1) — which defines Time Reversal Symmetry — **doesn't exist**. Consequently, we cannot derive this process in the same manner as described above. This implies that $R \circ \phi_t \circ R \circ \phi_t = I$ **does not hold true for these systems**.
>
> Thus, to enforce our model satisfy Time-Reversal Symmetry, we equivalently derive that two trajectories should converge based on $R \circ \phi_t \circ R \circ \phi_t = I$ form Lemma 1 and design our loss function based on this equivalent definition of time reversal symmetry.
>
> Though the form of the loss might have led to some misunderstanding that we are using reversibility, our loss function actually derived from the equivalent definition based on time reversal symmetry (from Lemma1). And in our previous response, we have explained that it differs from time reversibility, where the underlying dynamics of forward and backward trajectories can differ as long as the resulting trajectories are close. This should adequately clarify that our model's training is guided by time reversal symmetry.”
>
>
>
>
> >Q2 : About T
>
> W1: T was just for implementational usage. For instant, the time point of an ODE is 0,1,2,3,4,5, and the trajectory is $\hat{y}^{fwd}_i(0), \hat{y}^{fwd}_i(1),…\hat{y}^{fwd}_i(5)$.
>
> the -t would be 0,-1,-2,-3,-4,-5. Then we can use a T=5 to shift the timestamps from 0,-1,-2,-3,-4,-5 to 5,4,3,2,1,0. This kind of input facilitates us in using odeint to predict the reverse trajectory $\hat{y}^{rev}_i(5), \hat{y}^{rev}_i(4),…\hat{y}^{rev}_i(0)$.
>
> From the equivalent definition based on time reversal symmetry（in Lemma1), our We hope these two trajectory to be the same, so we derived our loss function as
> $\mathcal{L}^{reverse}=\sum_{i=1}^{N}\sum_{t=0}^{T}||{{\hat{y}}_i^{fwd}(t)-\{\hat{y}}^{rev}_i(T-t)}||_2^2$

---

> > ### Author Response · Authors · 2023-11-23
> >
> > Dear reviewers,
> >
> > We sincerely appreciate your valuable feedback. As the deadline for the author-reviewer discussion phase is approaching, we would like to check if our response addressed your concerns regarding the definition of time reversal symmetry. If there are any remaining issues or if you require further clarification, please feel free to inform us.
> >
> > Best Regards, Authors.

---

### Official Review · Reviewer_m559 · 2023-10-31

**Soundness:** 1 poor
**Presentation:** 2 fair
**Contribution:** 1 poor
**Rating:** 3
**Confidence:** 4

**Summary:**

The work contributes to Physically Informed NNs for the simulation of multibody/multiagent physical systems by introducing a loss regularization term that encourages reversible continuous-time trajectories in the neural feature space. The paper also provides a result showing that the regularized loss entails the minimization of higher order Taylor expansion terms in ODE integration.

**Strengths:**

The paper puts forward an interesting motivation when it claims that engaging with Hamiltonian-type neural models might pose too stringent requirements, both on energy preservation as well as on the learning model. The idea of focusing on symmetry/reversibility properties rather than on strict energy preservation requirements is interesting. Even more so if this can be coupled with an approach that is fairly general and computationally efficient to achieve (as it seems reasonable to assume with the proposed approach). The empirical results seems to confirm the claim (though with the limitation described in the weaknesses below).

**Weaknesses:**

W1) As highlighted in the “Strengths” part, the motivation of the work is compelling and the idea of addressing reversibility through a simple regularization term is interesting. Unfortunately these are not novel contributions of this paper. Rather they are adapted from Huh et al, NeurIPS 2020, essentially adding the graph NN dimension (which is straightforward extension) and the simplified reversibility regularization with associated theoretical analysis (which is a less straightforward one). Hence my issue here is that the work is perhaps a bit too incremental.

W2) Related with reversibility, I am under the impression that the paper is missing to position the work adequately with respect to some quite relevant related literature on the topic. The topic of reversible neural architectures has a good standing in some quite consolidated works which explicitly study the relationship with stability and non-dissipative diffusion [A]. At the same time, reversible graph flows are discussed when considering generative neural models [B]. More recently, few groups have been studying bracket-based dynamics as way to induce learning models (also on graphs) that mix reversible and irreversible behaviours  [C,D]. These are very relevant related works which deserve to be cited and confronted with both theoretically and empirically, especially as they take a different perspective of inducing provable reversibility as compared to “encouraged reversibility” as in this paper.

W3) The empirical analysis lacks sufficient details for reproducibility. I am missing in the main body (as well as in the appendices) a clear indication of the presence of a validation set, on which for instance one needs to identify the proper \alpha regularization weight. The empirical analysis itself is not very compelling as it is limited to few simple simulated physical systems. While this is somewhat consistent with some published earlier literature, in more recent works there is the tendency to validate models on more compelling and complex setups, e.g. involving simulation of MuJoCo dynamics or CMU human trajectory data.

[A] Bo Chang et al., Reversible Architectures for Arbitrarily Deep Residual Neural Networks, AAA 2018

[B] Jenny Liu et al, Graph Normalizing Flows, NeurIPS 2019

[C] K. Lee et al, Machine learning structure preserving brackets for forecasting irreversible processes, NeurIPS 2021

[D] A. Gruber et al, Reversible and irreversible bracket-based dynamics for deep graph neural networks, NeurIPS 2023

**Questions:**

Q1) Authors are invited to expand the empirical and theoretical comparison to other works related to reversible architectures (see the sample provided in the Weakness sections)

Q2) Can the Authors clarify the model selection setting of the work, in particular as pertains the optimization of the model hyperparameters?

---

> ### Author Response · Authors · 2023-11-21
> **Response to Reviewer m559 (Part I)**
>
> We thank you for your insightful comments on improving our paper. However, we believe there’s some misunderstanding and would like to make the following claims to address your concerns.
>
> >W1: The time-reversal regularization term heavily relies on ideas from Huh et al NeurIPS 2020, making its contributions appear incremental.
>
> A1:  Thanks for your comments. We would like to emphasize our contributions  1.) Our work is **the first** to establish the theoretical connection between the **physical meaning** of the time reversal symmetry loss with its **numerical benefits** in improving the accuracy in general, without requiring the systems to follow any energy constraint (Theorem 1). Therefore, we broaden the application of the proposed loss to more real-life dynamical systems. 2.) Secondly, time-reversal symmetry has different definitions that are equivalent. The loss functions in our paper and in Huh et.al. Neurips 2020 are built upon definitions in Lemma 1 and Eqn (5) respectively. Note that the loss functions are only approximations of time-reversal symmetry where they try to satisfy Lemma 1 and Eqn (5). We **theoretically** show in Appendix A.3 that our reversal loss is guaranteed to have a lower maximum error compared to their approximations. We also **empirically** show our loss is achieving better performance: In Table 1, $TANGO_{L_{rev}=\text{rev2}}$   is a model variant by replacing our reversal loss with the one proposed in Huh et al NeurIPS 2020. TANGO consistently outperforms $TANGO_{L_{rev}=\text{rev2}}$  across systems, showing the superiority of our approximation.
>
>
> >W2 & Q1: Discussion of relevant literature on reversible neural architectures and bracket-based dynamics.
>
> A2: Thanks for pointing out these important works. We will add discussions in the related work section to provide a more comprehensive background to the audience. However, we would like to claim that these papers are not directly comparable to our methods: a.) For AB, they are out of the scope of dynamical system modeling. The reversibility is about NN architectures in terms of layer depth, while we are interested in the time-reversal symmetry of physical dynamical systems. b.) For CD, they are both based on the inductive bias with the GENERIC formalism which is commonly seen in the nonequilibrium statistical mechanics literature, concerning the conservation of energy/entropy. However, the assumption that all dynamic systems fall under this inductive bias is not valid. Our work targets time-reversal symmetry for dynamical systems under classic mechanics, where the (ir)reversibility is about the trajectory of agents, instead of the system-wide quantity such as energy/entropy. We thank the reviewer for providing valuable insights from these papers and are happy to discuss future work building upon them to apply to other systems.

---

> ### Author Response · Authors · 2023-11-21
> **Response to Reviewer m559 (Part II)**
>
> >W3:  Experiment details and more datasets.
>
>
> A3: Thanks for your insightful suggestions. We will add the validation set details in Appendix D.4 where 10% of training samples are used as validation sets in our experiments. Towards the dataset, we use three different spring systems to illustrate the different combinations of energy conservation and time reversibility shown in Figure 1. We additionally use a challenging pendulum dataset to show our model performance is especially better compared to baselines. Although the data is simulated, it contains noise and is incompletely observed and irregularly sampled, to some extent reflecting the challenges that real-world data may face.
> For the MuJoCo human motion dataset, it differs from the simulated dataset in that it does not have explicit physical inductive bias. Since our work is positioned as physics-informed NN, time-reversal symmetry is not observed in MuJoCo so we believe our architecture design will not be very useful for it. We conducted experiments on the dataset from Huang.et.al. Neurips 2020 and showed the results below. we can see that TANGO outperforms existing baselines though the margin is relatively small.
>
> Table 1: Evaluation results on MSE ($10^{-2}$) on MoJoCo Dataset (Huang et.al. Neurips 2020)
>
> | $\text{Model \quad }$        | $\text{LatentODE \quad}$ | $\text{HODEN \quad}$  | $\text{TRS-ODEN}_\text{GNN} \quad$ | $\text{LGODE} \quad$   | $\text{TANGO} \quad$   |
> |--------------|-----------|----------|-------------------------|---------|---------|
> | MSE          | 2.9061     | 1.9855 |0.2609                  | 0.7610  | 0.2192  |
>
>
>
> >Q2:  The selection of model hyperparameters.
>
> A: TANGO follows the GraphODE architecture as in Huang et al. 2021, and we use the default architectural hyperparameters detailed in Appendix D.2. For training hyperparameters, we search from the following ranges: lr ∈ {1e-3, 5e-4, 1e-4, 5e-5, 1e-5, 5e-6}, batch_size ∈ {256, 512}. For spring datasets, we search $\alpha$ from a range of smaller values {0.01, 0.02, 0.1, 0.5, 1, 2, 5, 10}. For the Pendulum, we search $\alpha$ from a range of larger values {5, 10, 50, 100, 1000}.
>
> .

---

> > ### Comment · Reviewer_m559 · 2023-11-22
> > **Post rebuttal**
> >
> > I thank the Authors for their response: clarification on validation set are important and I appreciate the additional experiment on Mujoco. I would like to see the additional information being committed to a new version of the manuscript. Thanks also for confirming what are the points of novelties of your work. I am not however convinced by dismissing additional works A as irrelevant: the work use the reversibility concepts to provide indications on how to induce a certain inductive bias on NN architectures through an  interpretation of NN layers as discrete steps of an ODE. Nothing prevents from applying these concepts to an unfolded ODE implementing a physical systems (in its input/output map). I would nonetheless take these comments into consideration in the reviewers discussion and in the final score.

---

> ### Author Response · Authors · 2023-11-22
> **Clarification of Reversible NN against Time-Reversibility**
>
> Thanks reviewer for the acknowledgment of our response.
>
> We'd like to further clarify:
>
> In our response, we didn't claim reversible NN are **irrelevant** to our work. Instead, both of these works are motivated by the nature of dynamic system: reversible NN treat each RevNet unit as a transforming operator (which is reversible, i.e., input can be induced from output), and thus its forward pass can be treated as ODE without storing past activations. Definitely this line of work can be used to replace neural operator in any Neural-ODE for dynamic system modelling.
>
> However, we'd like to clarify that simply replacing with a RevNet didn't resolve the time-reversible problem that we're studying. The core of RevNet is that input can be recovered from output via a reversible operation (which is another operator), similar as any linear operator $W(\cdot)$ have a reversed projector $W^{-1}(\cdot)$. In contrary, what we want to study is that the **same operator** can be used for both forward and backward prediction over time, and keep the trajectory same. (see our **eq 6 and 8, we are using the same $g(\cdot)$, instead of $g^{-1}(\cdot)$**, and we don't need and assume operator $g(\cdot)$ to be reversible, instead, we expect it can predict same trajectory when predicting the backward via reversing the time)
>
> In summary, though both reversible NN and time-reversal share similar insights and intuition, they're talking about very different thing: reversible NN make every operator $g(\cdot)$ having a $g^{-1}(\cdot)$, while time-reversible assume the trajectory get from $\hat{z}^{fwd}=g(z)$ and $\hat{z}^{bwd}=-g(z)$ to be closer. **Making $g$ to be reversible cannot make the system to be time-reversible.** We totally agree they are orthogonal to each other and very likely combining them together can achieve better performance, which we might leave for future exploration. We will add the above discussion into our paper.
>
>
> We have added the discussion in our revised draft and hope our clarification resolves your concern.  Please let us know if you have further concerns and questions

---

> > ### Author Response · Authors · 2023-11-23
> >
> > Dear reviewers,
> >
> > We sincerely appreciate your valuable feedback. As the deadline for the author-reviewer discussion phase is approaching, we would like to check if our response addressed your concerns regarding the reversible NN. If there are any remaining issues or if you require further clarification, please feel free to inform us.
> >
> > Best Regards,
> > Authors.

---

### Author Response · Authors · 2023-11-22
**General Response**

We sincerely thank all reviewers and the program chairs for their valuable time and insightful feedback. Based on the discussion, we have revised our manuscript to improve clarification (see changes in red).  We hope the current draft is able to address your concerns.

---

### Meta-Review · Area_Chair_d8Z5 · 2023-12-10

**Metareview:**

The paper proposes a self-supervised regularization term to inject time-reversal symmetry as a strong inductive bias for learning complex multi-agent system dynamics from data. This regularization aligns the forward and backward trajectories predicted by a continuous graph neural network-based ordinary differential equation (GraphODE) and minimizes higher-order Taylor expansion terms during the ODE integration steps, making the model more noise-tolerant and applicable to irreversible systems. Experimental results show that the proposed method achieves an MSE improvement of 11.5% on a challenging chaotic triple-pendulum system.

Firstly, the studied topic is an interesting one, which focuses on symmetry/reversibility properties rather than on strict energy preservation requirements. Secondly, the experiments show TANGO outperforms many baselines, including Latent NODEs, Hamiltonian NODEs, TRS NODEs, and graph NODEs on a range of problems. Finally, the paper is well organized, and presents a detailed literature review.

Firstly, the novelty of the paper is limited, since many components are adapted from existing works. Secondly, the compared baselines should be selected more carefully, since the current ones cannot fully support the claims. Finally, the paper should be polished more carefully, such as comparison with reversible neural architectures, providing a more rigorous distinction between time-reversal symmetry and time reversibility (invertibility), and providing standard deviations of experiments for checking statistical significance.

**Justification For Why Not Higher Score:**

The novelty of the paper is limited, and  the compared baselines should be selected more carefully.

**Justification For Why Not Lower Score:**

N/A

---

### Decision · Program_Chairs · 2024-01-16

Reject